# The dynamics of mutational selection in cutaneous squamous carcinogenesis
Greta Skrupskelyte [1], Joanna C. Fowler [2], Stefan Dentro[2,3,4], Carine Winkler [5], Irina Abnizova[2], Niklas Beumer [3,6,7], Roshan Sood [2], Thomas Quarrell[8], Charlotte King[2], Jivko Kamarashev[9], Emmanuella Guenova[10,12], Moritz Gerstung [3,4], Benjamin A. Hall [11], Liliane Michalik[5] & Philip H. Jones [2,13] ✉

Identifying the mutant genes that are selected during carcinogenesis is key to identifying candidates for intervention and understanding the processes that promote transformation. Here we applied two selection metrics to study the dynamics of mutational selection in a mouse model of ultraviolet light driven skin carcinogenesis in which multiple synchronous tumors develop in each animal. Sequencing normal skin and tumors over a time course revealed two genetic routes to squamous carcinoma. Nonsynonymous *Trp53* mutants were positively selected in both epidermis and tumors and present in 90% of tumors. The remaining tumors carried other oncogenic mutants, including activating *Kras* mutations. However, other positively selected mutant genes lost their competitive advantage in heavily mutated epidermis and in tumors. We found ten mutant genes under negative selection in normal skin, one of which was also negatively selected in tumors. In addition one gene was negatively selected in tumors but not normal skin. We conclude that analysing selection in normal tissue alongside tumors may resolve the dynamics of selection in carcinogenesis and refine the identification of cancer drivers.

Mutant selection and tumor evolution may be inferred retrospectively from cancer genomes[1–3]. However, ageing normal tissues, such as the skin epidermis, develop into a patchwork of positively selected mutant clones, including mutants that occur recurrently in cancer[4–15]. Some mutants that drive clonal expansions in normal tissues are oncogenic, but others may be neutral or even inhibit carcinogenesis[1,5,16,17]. Furthermore, the altered environment within growing tumors may lead to the selection of mutants that are neutral in normal tissues[1,16]. Most tumor sequencing studies have not analysed the normal tissue from which the tumor arose, so that the oncogenicity of some mutant genes selected in tumors remains unclear.

As well as positively selected mutants, identifying mutant genes that are negatively selected, particularly those negatively selected in tumors, may reveal therapeutic targets[18]. The existence of negative selection in human

tumors was initially controversial, but more recent evidence argues it is widespread[2,18–20]. Many studies of normal tissues have too few synonymous mutations to identify negatively selected genes[2]. However, those studies with sufficient power have detected negative selection of somatic mutants in normal skin and oral epithelium[5,21].

Studying selection in humans is challenging due to variations in the germline, environmental exposures and the age at which lesions develop and the need to sequence both tumor and normal tissue[5,10,22,23]. These factors may be mitigated in mouse models in which multiple, synchronous lesions develop, facilitating comparison of normal tissue and tumors in the same animal. Here, we used the well-characterized SKH-1 mouse model of UV light-induced skin cancer to explore the dynamics of selection in squamous carcinogenesis[24–27]. A key feature of these animals is a high susceptibility to

¹Cambridge Stem cell Institute, University of Cambridge, Cambridge, UK. ²Wellcome Sanger Institute, Wellcome Genome Campus, Hinxton, Cambridge, UK. ³European Molecular Biology Laboratory, European Bioinformatics Institute, Cambridge, UK. ⁴Artificial Intelligence in Oncology (B450), Deutsches Krebs-forschungszentrum, Heidelberg, Germany. ⁵Centre for Integrative Genomics, Faculty of Biology and Medicine, University of Lausanne, Lausanne, Switzerland. ⁶Institute of Immunology, University Medical Centre Mainz, Mainz, Germany. ⁷Research Centre for Immunotherapy, University Medical Centre Mainz, Mainz, Germany. ⁸School of Clinical Medicine, Addenbrooke's Hospital, Hills Rd, Cambridge, UK. ⁹Department of Dermatology, UniversitätsSpital, Zurich, Switzerland. ¹⁰Clinical Department of Immunodermatology, Kepler University Clinic and Medical Faculty, Johannes Kepler University, Mainz, Austria. ¹¹Dept of Med Physics & Biomedical Engineering, University College London, London, UK. ¹²Clinical Research Institute for Inflammation Medicine, Medical Faculty, Johannes Kepler University, Linz, Austria. ¹³Department of Oncology, University of Cambridge, Hutchinson Research Centre, Cambridge Biomedical Campus, Cambridge, UK. ✉e-mail: pj3@sanger.ac.uk

UV-induced tumor formation, which is dependent on a germline mutation in the *Hr* gene[27,28]. Repeated exposure to UV light results in skin thickening followed by the development of multiple tumors per animal, which have pathologic features of premalignant actinic keratosis and invasive squamous cell carcinoma[24,29]. Exome sequencing of SKH-1 cutaneous SCC (cSCC) reveals protein-altering *Trp53* mutations are present in most tumors, thus resembling human skin sSCC[22,24,30–32].

We applied two metrics of selection to normal and tumor tissue. The first was the widely used approach of estimating the ratio of the number of protein-altering (missense, nonsense, essential splice, and insertions/deletions) to synonymous mutations, dN/dS, within each gene, so that for missense mutations, for example,

$$dN/dS_{mis} = n\_missense/n\_synonymous$$

where n is the total number of mutations of each type across the gene. The implementation of dN/dS used, dNdScv, controls for trinucleotide mutational signatures, sequence composition, and variation in mutation rates between genes, to avoid the known biases in dN/dS estimation[2,7]. A normalized dN/dS value of 1 is neutral; values significantly above 1 after multiple test correction indicate positive selection, while values below 1 indicate negative selection. In this experiment, dN/dS values reflect the likelihood that a nonsynonymous mutant clone will reach a detectable clone size compared with a synonymous mutant clone in the same gene.

A second measure of selection that may be suitably applied to the current analysis of normal tissue and tumors from a single cell lineage is the ratio of variant allele frequencies (VAF, defined as the proportion of sequencing reads reporting a mutation within each sample) of nonsynonymous and synonymous mutants in each gene (VAF_ns/VAF_s).

$$VAF\_ns/VAF\_S = \text{sum of VAF of all nonsynonymous mutants}/$$
$$\text{sum of all synonymous mutants in each gene.}$$

Unlike dN/dS, the VAF of all types of nonsynonymous mutations is pooled in the calculation of VAF_ns. If a mutant gene drives clonal expansion, the VAF of nonsynonymous mutants will be significantly larger than the VAF of synonymous mutants in the same gene due to an increased proportion of nonsynonymous mutant cells in the sample, giving a VAF_ns/VAF_S ratio significantly above 1. In contrast, for neutral "passenger" mutations, the VAF_ns/VAF_s ratio will be close to 1.

Here, we assayed the mutational landscape of normal skin epidermis and tumors from a cohort of UV-exposed SKH-1 mice in a time course to explore the dynamics of mutant selection. Two methods of sequencing were employed. To detect mutant clones, we used targeted sequencing of DNA extracted from small samples of normal tissue, where the presence of mutant clones is revealed by mutant reads for each sequenced gene. As mutant clones may only be present in a small fraction of each tissue sample, we used an algorithm, ShearwaterML, to detect mutations. This builds a statistical model of background sequencing errors at each base to identify true variants which have a statistical excess of mutant reads, allowing the detection of mutations present in as few as 1% of the cells in a sample[5,7,33]. We also measured the mutational burden, the number of mutations per megabase across the genome, using a method called Nanoseq. This is a modification of duplex sequencing, which allows the detection of somatic mutations across a third of the genome with an error rate <5/10^9 base pairs[34]. Nanoseq can detect a mutation even if it occurs in just a single DNA molecule in a sample. Importantly for this experiment, Nanoseq estimates of mutational burden are not affected by the expansion of mutant clones.

We found a highly competitive and dynamic mutational landscape during carcinogenesis with two evolutionary trajectories of cSCC formation. Protein-altering *Trp53* mutants were under strong positive selection in both normal tissue and tumors. In contrast, other nonsynonymous mutant genes, which are common in human cSCC, drive clonal expansion in normal tissue

but appear not to do so within tumors. Negative selection was attenuated in tumors compared with normal tissue.

## Results

### UV-exposed SKH-1 mouse epidermis becomes densely mutated

A cohort of 20 SKH-1 mice were exposed to three sub-minimal erythema (below sunburn) doses of UV radiation per week. Animals were culled after 8 to 27 weeks of UV irradiation of the back (dorsal) skin (Fig. 1a and Supplementary Table 1). Epidermal histology confirmed the expected increase in epidermal thickness in the UV-exposed dorsal skin (Supplementary Fig. 1a, b)[29]. The skin was collected, and tumors were dissected. Part of each lesion was removed for histology, after which spaced cryosections from the remaining tumor were imaged and sequenced (Methods). Wholemounts of normal epidermis from dorsal and ventral skin were then prepared, imaged to confirm the absence of tumors and sent for targeted sequencing to a median depth of 931x coverage for 74 genes associated with squamous cancer (Supplementary Fig. 1c, d and Supplementary Data 1, 2).

We began by analysing the genomic alterations in normal skin over time using Nanoseq. The mutational burden in dorsal skin rose progressively, reaching a median of over 30 single-base substitutions (SBS)/Megabase by 24 weeks (Fig. 1b and Supplementary Data 3). The mutational burden estimates by NanoSeq were 20 SBS/Mb higher than those derived from targeted sequencing (Supplementary Fig. 1e). For comparison, the median mutational burden in histologically normal, sun-exposed, middle aged, human skin estimated from targeted sequencing typically ranges from 10–20 SBS/Mb, similar to the burden at the 8-week time point (Supplementary Fig. 1f), but may be several fold higher in patients with multiple keratinocyte cancers[5,11,12].

The mutational spectrum was dominated by C > T transitions, consistent with UV exposure, in both targeted and Nanoseq sequencing of the epidermis (Supplementary Fig. 2a, b and Supplementary Data 4)[7]. The mutational spectra from targeted and Nanoseq sequencing were highly correlated (Supplementary Fig. 2c). Double Base Substitutions (DBS) also increased with UV exposure and 35% of the changes were CC > TT, in line with previous observations (Fig. 1c). The ventral epidermis, which was not directly exposed to UV light, remained sparsely mutated throughout the experiment (Supplementary Fig. 1h). Non-UV exposed control mice also showed very few mutations (Supplementary Data 5).

Accompanying the changes in mutational burden we detected a progressive increase in the number of mutations per mm² of dorsal epidermis with the duration of irradiation (Fig. 1d). Across all time points, the total number of SBS, DBS and indels detected in the dorsal epidermis was 94138, 2126, and 471 respectively in a total area of 10.26 cm² epidermis (Supplementary Data 5).

### Selection of mutant genes in UV-exposed epidermis

To explore the selection of mutant genes in the epidermis, we first determined the dN/dS ratio for each sequenced gene[7,35]. As missense dN/dS ratios may be misleading if both gain and loss of function mutants occur in the same gene, we only considered the dN/dS for nonsense and essential splice mutants for determining which mutant genes were under selection[5]. After correction for mutational spectrum, genomic context and multiple testing, eight mutant genes, including *Trp53*, *Notch1*, *Notch2*, *Fat1*, and *Ajuba* emerged as being under positive selection in the epidermis (dNdS_nonsense/ essential splice >1.3, q < 0.01) (Fig. 2a and Supplementary Data 6). The human equivalents of these mutant genes are positively selected in sun-exposed human epidermis[5–7]. Mutants of the skin tumor suppressor histone lysine methyltransferase *Kmt2c*, atypical cadherin *Fat2*, and the tyrosine kinase insulin receptor family gene *Ros1* were also under positive selection (Fig. 2a)[36].

The large number of mutants enabled us to detect negatively selected mutant genes. A dN/dS analysis revealed ten genes under negative selection in epidermis (dN/dS_nonsense/essential splice <0.6, q < 0.01, Fig. 2b and Supplementary Data 6). These included inactivating mutations of the growth factor receptors *Egfr*, *Erbb2*, *Erbb3*, and their downstream signaling

intermediates *Mtor* and *Pik3ca* in keeping with the pro-proliferative function of the wild-type genes in the epidermis[37–39]. Mutations disrupting the epigenetic regulators *Ep300* and *Smarca4*, the latter essential for epidermal differentiation, the Hedgehog pathway regulator *Smo*, *Casp8* which is required for epidermal homeostasis, and *Atp2a2*, mutation of which causes ER stress and the human skin disorder Darier's disease were also negatively selected[40–44].

Next, we analysed the VAF of the dN/dS positive genes. At all time points, Vaf_ns/Vaf_s of *Trp53* mutant clones was significantly above 1 (Fig. 2c and Supplementary Data 7). However, this was not the case for other positively selected mutants. For *Notch1*, Vaf_ns/Vaf_s was significantly above 1 at 8 weeks but not at later time points (Fig. 2d). For all other positively selected mutant gene clones, Vaf_ns/Vaf_s was consistent with neutrality at every time point (Fig. 2e, Supplementary Fig. 3a–e, and Supplementary Data 7).

Finally, we investigated the proportion of epidermis colonized by positively selected mutant genes. This may be estimated by summing the VAF for all protein-altering mutations in each gene at each time point. In a diploid tissue, this summed VAF is proportional to the area of epidermis occupied by mutant cells (Fig. 2f–h, Supplementary Fig. 3f–k, and Supplementary Data 8). By 16 weeks UV exposure most of the epidermis was taken over by mutations in genes under positive selection (Supplementary Fig. 3f). Of these genes, mutant *Trp53* colonized epidermis more rapidly and extensively than any other positively selected gene, occupying a third of the area of the skin after 16 weeks of UV exposure (Fig. 2f). *Notch1* mutants eventually colonize a quarter of the epidermis, whilst *Fat1* and other mutant genes were less prevalent (Fig. 2g, h, Supplementary Fig. 3g–k, and Supplementary Data 8).

## Characterization of tumor genomes

Turning to the tumors, we characterized the lesions which developed from 16 weeks of UV exposure onwards (Fig. 3a). By 27 weeks, animals had between 3 and 12 lesions each. In total, 150 tumors were dissected, with part sent for histopathological examination and the remainder frozen. Histology identified 22 premalignant actinic keratoses (AK) and 126 cSCC in the dorsal skin. cSCC were graded into three categories, I–III, based on their pathological characteristics, with I being well differentiated and III the most malignant (Methods, Supplementary Fig. 3a–c, and Supplementary Data 9). Thirteen AK and 66 cSCC were large enough for multiregional sequencing, using three or more spaced cryosections from which the tumor was micodissected. The size and location of lesions is shown in Supplementary Fig. 3c, d. Targeted DNA sequencing was performed to a median depth of 926x coverage (Supplementary Fig. 3e Methods and Supplementary Data 2).

The total number of SBS, DBS, and indels detected in the 79 sequenced lesions was 49357, 1513, and 235, respectively (Supplementary Data 10). The mutational burden estimated from targeted sequencing data were similar in AK and all grades of cSCC ($p = 0.55$, Kruskal–Wallis test) but rose with the duration of UV exposure (Fig. 3b, c). The proportion of the genome with copy number alterations (CNA), estimated from off-target reads, also rose with the length of UV exposure (Fig. 3d, Methods, and Supplementary Figs. 4, 5). CNA changes were widely distributed over chromosomes, though gains were most frequent on chromosomes 2 and 5 and losses on chromosome 1 (Supplementary Fig. 3f, g and Supplementary Data 11).

## Mutational selection in tumors

We next investigated mutant selection within tumors, beginning with dN/dS analysis. Positively selected mutant genes in tumors were the same as in the epidermis, except that *Kmt2c* was not selected in tumors (Figs. 2a, 4a). However, negative selection was markedly different. Only nonsynonymous mutants of *Egfr* and nonsense/essential splice and indel mutants of *Rb1*, which was not under selection in the epidermis, were under negative selection. (Fig. 4b and Supplementary Data 12).

Vaf_ns/Vaf_s analysis showed only mutant *Trp53* had Vaf_ns/Vaf_s significantly above 1 (Fig. 4c–e and Supplemental Data 13). Consistent with this, the proportion of tumor tissue mutant for mutant *Trp53* rose

progressively and was almost double that of other mutant genes under positive selection (Fig. 4f–h and Supplemental Data 14). Most *Trp53* missense mutants resulted in codon changes within the DNA binding domain in both epidermis and tumors (Supplementary Fig. 7). A similar distribution of *TP53* missense mutants is seen in human sun-exposed skin and cutaneous squamous cell carcinoma[5,7,23,45].

We noted there were eight tumors from eight mice where the proportion of *Trp53* mutants was below that of the epidermis (Fig. 5a, Supplementary Fig. 8, and Supplementary Data 15). Comparison of selection in these *Trp53low* tumors compared with *Trp53high* lesions revealed differences in the selection of other mutant genes, including stronger selection of *Kras* and *Tgfbr1* mutants in the *Trp53low* group. Conversely, there were no differences in mutant gene selection comparing tumors with a proportion of *Notch1* or *Fat1* mutant cells above or below that in the epidermis (Supplementary Fig. 9).

Three *Trp53low* tumors harbored gain-of-function mutant *Kras* (one each of lesions with $Kras^{G12D}$, $Kras^{G12C}$, and $Kras^{P34L}$) and two were mutant for *Tgfbr1* (one with clonal nonsense and missense mutations and the other with a missense mutation). (Fig. 5b, c and Supplementary Fig. 8). Functional studies in mice argue gain-of-function *Kras* and *Tgfbr1* mutations promote squamous skin tumor formation[46–49]. *Notch2* and *Kmt2c* mutations, found recurrently in mouse and human skin SCC, were present in two tumors[50–53]. In one tumor, there were no detected mutations in the panel of sequenced genes.

Mutant *Trp53*-high lesions were more likely to undergo CNA than mutant *Trp53*-low tumors (Fig. 5d, Methods). These results suggest there may be two routes of cSCC evolution, with most lesions being *Trp53*-high and a minority *Trp53-low*, harboring alternative driver mutants, including mutant *Kras* and *Tgfbr1*.

An advantage of the SKH-1 model is that the multiplicity of tumors in each animal allows investigation of whether genome changes are more similar for lesions arising within the same host than across different mice. We focused on animals at 24 and 27 weeks, where the number of tumors per mouse is highest. There was no significant difference in the proportion of *Trp53* mutant cells (Fig. 6a, b), mutational burden (Fig. 6c, d), or the proportion of copy number altered genome (Fig. 6e, f) between tumors in different animals at the same time point (two-tailed *F* test). The variation within each animal in each of these three parameters was similar to that between animals. Animals developed both AK and SCC, and there was no significant difference in histological tumor types between each animal (Fig. 6g, h, two-tailed *F* test). We conclude that tumor genomes were not influenced by host factors in SKH-1 mice.

## Discussion

The mutational landscape of UV-exposed SKH-1 mouse epidermis was comparable to that in humans. By the end of the experiment, the mutational burden of normal skin had risen to a similar level as that of sun-exposed skin in humans with a history of multiple keratinocyte cancer[12]. The mutational spectrum in mouse epidermis was narrower than in human skin, being almost exclusively confined to UV-associated C > T and CC > TT changes. In contrast, in human epidermis, a third or more of mutations are due to cell intrinsic processes (COSMIC signatures SBS1, 5, and 40), resulting in a much broader mutational spectrum[5,23]. This difference is likely to reflect ageing in humans as the burden of mutations due to these "clock-like" signatures increases in proportion with time[54].

dN/dS analysis showed the mutant genes under positive selection in UV-treated mouse epidermis overlapped with those positively selected in human sun-exposed skin (Fig. 7)[5,7,23]. *Trp53* was the fittest positively selected mutant in mouse epidermis, evidenced by VAF_ns/VAF_s values significantly above 1 throughout the experiment. In contrast, for the other positively selected mutant genes VAF_ns/VAF_s was significantly above neutrality only for *Notch1* at the 8-week time point. How might the discrepancy between a positive a dN/dS ratio and the neutral VAF_ns/VAF_s be explained? Lineage tracing studies in squamous epithelia in transgenic mice show that mutants such as *Trp53* and *Notch1* expand rapidly for a

short period while mutants are competing with less fit wild-type cells[17,55–57]. However, once mutant cells are surrounded by cells of similar fitness, their phenotype and competitive fitness revert towards wild type. It is this phenotypic reversion that enables tissues like the epidermis to carry a high proportion of mutant cells and yet continue to function normally. The positive selection detected by dN/dS may reflect the brief period of clonal expansion in a wild-type background whereas neutral VAF_ns/VAF_s result from a much longer period of neutral dynamics following phenotypic

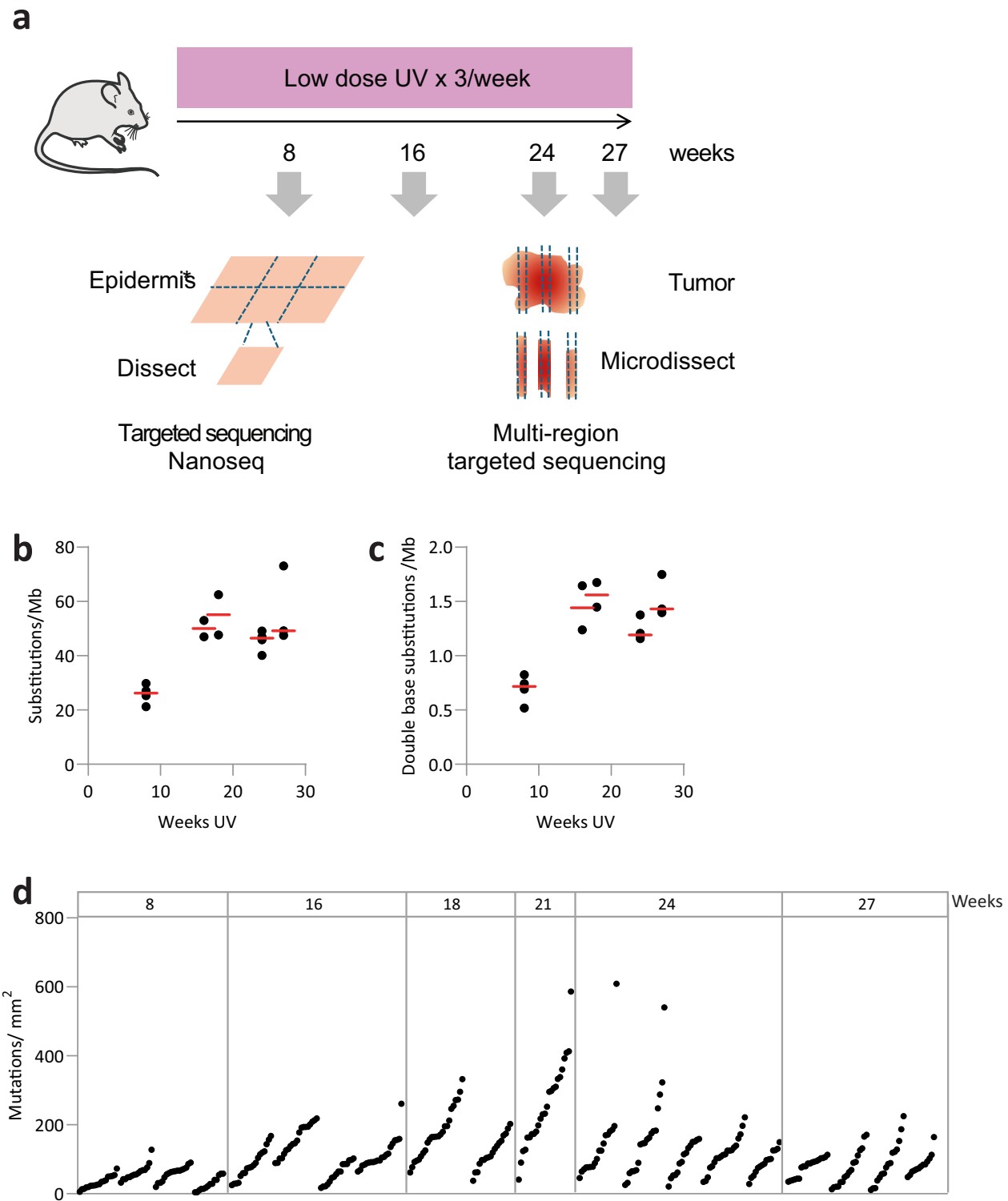

**Fig. 1 | UV induced mutations in epidermis. a** Protocol. Following sub-minimal erythema UV dose epidermis and tumors are dissected and sequenced as shown. **b, c** Mutational burden in irradiated dorsal epidermis. **b** Substitutions/Megabase, **c** double base substitutions/megabase, both determined by NanoSeq. Each dot represents one mouse, red bar indicates the median. Source Data: Supplementary Data 3 **d** Density of mutations, each dot is a different epidermal grid sample, grouped by mouse. Source data: Supplementary Data 3, 5.

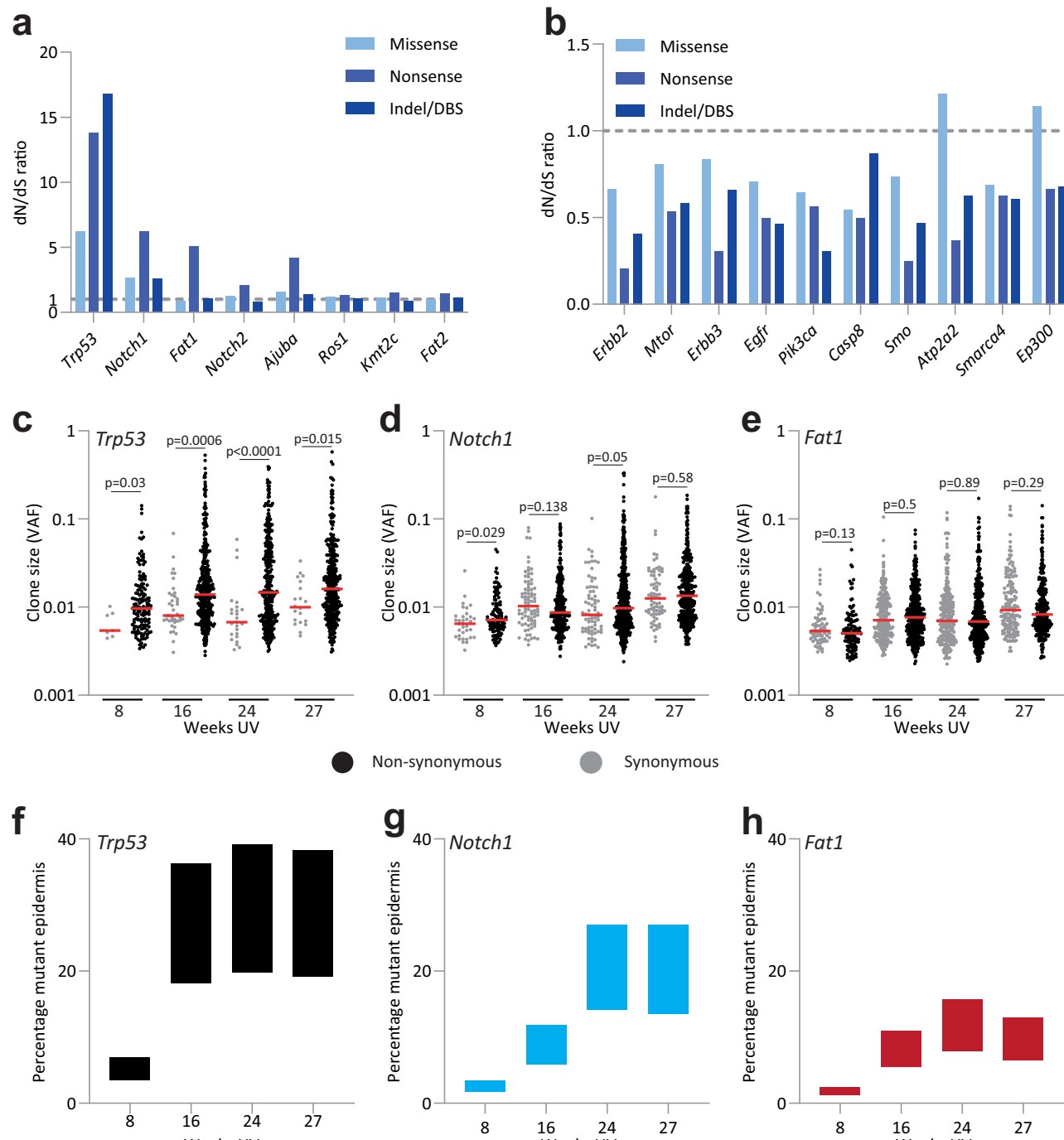

**Fig. 2 | Positive and negative selection of mutant genes in the dorsal epidermis. a, b** Nonsynonymous: synonymous (dN/dS) ratios in dorsal skin. Mutations in all mice across all time points are pooled; 1 is neutral. **a** Genes under positive selection ($q < 0.01$ and dNdS ratio >1.3 for nonsense/splice mutations). **b** Negatively selected genes ($q < 0.01$ and dN/dS ratio <0.6 for nonsense/ splice mutations). Source Data: Supplementary Data 6. **c–e** Nonsynonymous (black points) and synonymous (gray points) VAF distribution for *Trp53* (**c**), *Notch1* (**d**), and *Fat1* (**e**), p values determined by two-tailed Wilcoxon test. Red lines indicate the median. Source Data: Supplementary Data 7. **f–h** Percentage area of dorsal epidermis colonized by cells mutant for *Trp53* (**f**), *Notch1* (**g**), and *Fat1* (**h**), estimated from summed variant allele frequency at each time point. Data shows the average across all mice for each time point, $n = 4$ mice for each time point. Upper and lower bounds reflect uncertainty in copy number and multiple mutations per cell. Source Data: Supplementary Data 8.

reversion. As the tissue approaches saturation with positively selected mutants at 16 weeks, only *Trp53* mutants have a robustly positive VAF_ns/ VAF_s. Human *TP53* mutants are more strongly selected than other mutants in human facial skin which is saturated with mutant clones, arguing they have a higher fitness than other positively selected mutants[23]. By contrast, in facial skin which is sparsely mutated and has a lower burden of UV-induced mutations, *NOTCH1* and *NOTCH2* mutants are preferentially

selected, and *TP53* mutants are comparatively rare[23]. The high fitness of mutant *Trp53* may result from continuing UV light exposure, which drives the exponential expansion of *p53* mutant clones in mouse and human skin[57–59].

Four mutant genes, *Fat2, Ajuba, Ros1* and *Kmt2c* were positively selected in mice but have not been shown to be selected in human epidermis. Of these genes, *Ros1*-deleted cells have been reported to drive clonal

expansion in a CRISPR/*Cas9* screen in normal mouse epidermis. *Kmt2c* was identified as a tumor suppressor in a mouse transposon screen for genes regulating squamous carcinogenesis, and the related histone lysine methyltransferase, KMT2D, is selected in human epidermis[36,60]. *AJUBA*, which regulates keratinocyte cytoskeletal signaling via RAC, is positively selected in other human squamous epithelia[21,35,61]. Of the ten negatively selected mutants in mouse, only inactivating *PIK3CA* mutants have been shown to be negatively selected in human skin[5]. This may reflect the smaller number of mutations detected in human studies, as sufficient synonymous mutations are required to identify negatively selected genes.

Turning to carcinogenesis, the data were consistent with two convergent evolutionary routes to develop cSCC, both of which may be followed in the same animal. In both SKH-1 mice and humans, the majority of UV-induced tumors are *p53* mutant[8,22,24,49,50,62–66]. However, *p53* wild-type tumors that harbor other drivers, including oncogenic mutant *Kras* and *Tfbr1,* were also seen, paralleling *TP53* wild-type human tumors[8,22,49,66,67]. Copy number alterations were more likely to occur in *TP53* mutant than wild-type tumors, a finding also seen in human cSCC[22].

The dN/dS positive mutant genes were similar in the epidermis and tumors (Figs. 2a, 4a). This is also the case for the commonest positively selected mutant genes in sun-exposed human skin, *NOTCH1, NOTCH2, TP53,* and *FAT1,* which are also frequently mutated in AK and cSCC[5,7,22,31,68]. However, this need not mean that the positively selected mutants are driving tumor initiation and growth[17,69]. Mutations that are selected in the epidermis are likely to be found in a subset of tumors, even if they do not contribute to transformation[5,7]. In tumors, Vaf_ns/VAf_s is positive for *Trp53*, suggesting protein-altering *Trp53* mutations may drive tumor growth (Fig. 4f). However, this is not the case for other positively selected mutants for which Vaf_ns/VAf_s is neutral (Supplemental Data 13). Mutant genes in tumors that do not drive tumor growth may contribute to orthogonal malignant phenotypes such as invasion, epithelial-mesenchymal transition and metastasis[30,51,70].

There are no comparable human studies in which mutations in cSCC and AK have been directly compared with adjacent skin. Mutants that drive clonal expansions in normal tissues or premalignant lesions may be neutral or even inhibitory in malignant transformation[5,17,35,69,71]. In humans, the proportion of *TP53* mutant tumors is increased compared with the fraction of *TP53* mutant sun-exposed skin[5]. Conversely, the proportion of *NOTCH1* and *FAT1* mutant human tumors is similar to the proportion of normal skin mutants for these genes[5]. Collectively these observations suggest *NOTCH1* and *FAT1* may not make a substantial contribution to the transformation and growth of cSCC.

Negative selection was markedly different between normal tissue and tumors. In the epidermis, most of the negatively selected mutants are known to impact keratinocyte proliferation or differentiation. In tumors, negative selection was attenuated, and only two negatively selected mutant genes were found. The reasons for this may be that mutants with a negative fitness impact hitchhike along with positively selected mutations, so the net fitness of the clone remains positive. Such 'genomic hitchhiking' is most likely to occur in lesions with a high mutational burden, such as those in this study[20]. In addition, the degree of spatial competition between clones in an expanding lesion is reduced compared with the "zero sum game" of the epidermis, and environmental differences between tumor and epidermis may contribute to altered selection[32,57]. Mutant *Egfr* was negatively selected in both epidermis and tumors, consistent with the impairment of mouse skin tumor growth following the transgenic deletion of *Efgr* and EGFR inhibitor studies in human skin SCC[72,73] (Fig. 7). The negative selection of *Egfr* in normal epidermis reveals a dependency that correlates with the skin toxicity of EGFR inhibitors in humans[74,75]. The negative selection of non-sense and essential splice *Rb1* mutants in tumors seems at odds with the tumor suppressor role of RB1 in a range of human cancers[76]. However, this finding is consistent with the results of conditional *Rb1* deletion in the epidermis of adult *Rb1*[flox/flox] mice[77]. In a chemical carcinogenesis protocol, *Rb1* loss resulted in fewer and smaller tumors compared with wild-type controls[77]. The reduced size of the *Rb1*[-/-] tumors is due to increased apoptosis

in tumor cells, consistent with negative selection of Rb1 in mouse skin tumors[77].

Limitations of the study follow from the targeted sequencing approach used. There may be additional positively selected mutants in genes not in the sequenced panel, and universal essential and other genes under negative selection were not included. Furthermore, the resolution of the approach used to identify CNA is low, and changes smaller than a chromosome arm amplification or deletion may not be detected. The estimate of the proportion of the genome with CNA should be viewed as a lower bound. Sample size is also an issue, with only 8 *Trp53* low lesions. The data were consistent with there being two routes of tumor evolution, *Trp53high* and *Trp53low*, but a larger sample of lesions would be required to confirm this statistically.

We conclude that the genomic evolution of UV-exposed normal skin and the tumors that arise from it is convergent in mice and humans. Both positive and negative selection varies between normal tissue and tumors. Alterations in negative selection between tumors and normal tissue may be a general feature of carcinogenesis. Mutant genes negatively selected in both normal tissue and tumors suggest toxicity may occur in the normal epidermis if the gene targeted therapeutically. Conversely, mutant genes under negative selection only in tumors may be optimal candidates for antitumor therapy[78]. Model systems such as that employed here offer a route to resolving the dynamics of selection in carcinogenesis. Finally, this work demonstrates the utility of sequencing normal tissue alongside tumors in the same individuals.

## Methods

### Husbandry
A total of 52 SKH-1-hr hairless immunocompetent mice (Crl:SKH-1-Hr[hr], Charles River Laboratories) were housed in a standard colony (4–5 animals per cage), in a temperature- and light-controlled environment (12/12-h light/dark cycle) and maintained with water and food ad libitum.

### UVB-irradiation
Mice were UV irradiated on their backs three times per week with 70 mJ/cm[2] of UVB using a GL40E 40 W tube (SNEE), which emits most of its energy within the UVB range (90%; emission spectrum 280–370; 10% UVA)[79]. Doses of UVB (312 nm) were monitored using an appropriate radiometer. Non-irradiated (sham-manipulated) aged-matched mice were used as controls. Each mouse was an independent experimental unit. Sample size was estimated from previous reports[24,25]. Numbers of animals and tumors per time point are shown in Supplementary Table 1.

The time of tumor appearance, tumor number and size were monitored three times per week. Euthanasia (using $CO_2$ gas in accordance with the requirements of the Veterinary Office of the Canton Vaud (Switzerland) and Federal Swiss Veterinary Office Guidelines). The Veterinary Office of the Canton Vaud permitted a maximum tumor size of 0.5 cm[3] and this limit was not exceeded in any experiments. The maximum permitted weight loss was 10% of the initial weight, and this limit was not exceeded in any experiments. Tumors, non-tumoural dorsal skin and ventral skin samples were collected.

### Tumor grading
Tumors were graded as previously described[79]. Mouse skin tumors were paraformaldehyde-fixed and paraffin-embedded. Tissue sections (4 μm) were stained with hematoxylin/eosin. Histological analysis of actinic keratosis and tumor classification were performed in a blind manner by a dermatopathologist.

SCCs were classified according to the Broders' classification based on the degree of SCC keratinization and of keratinocyte differentiation[80]. This classification is as follows: SCC Grade I: 75% keratinocytes are well differentiated; SCC Grade II: >50% keratinocytes are well differentiated; SCC Grade III: >25% keratinocytes are well differentiated, and SCC Grade 4: <25% keratinocytes are well differentiated. Actinic keratoses were defined histologically and classified as grade I, II, or III based on the degree of

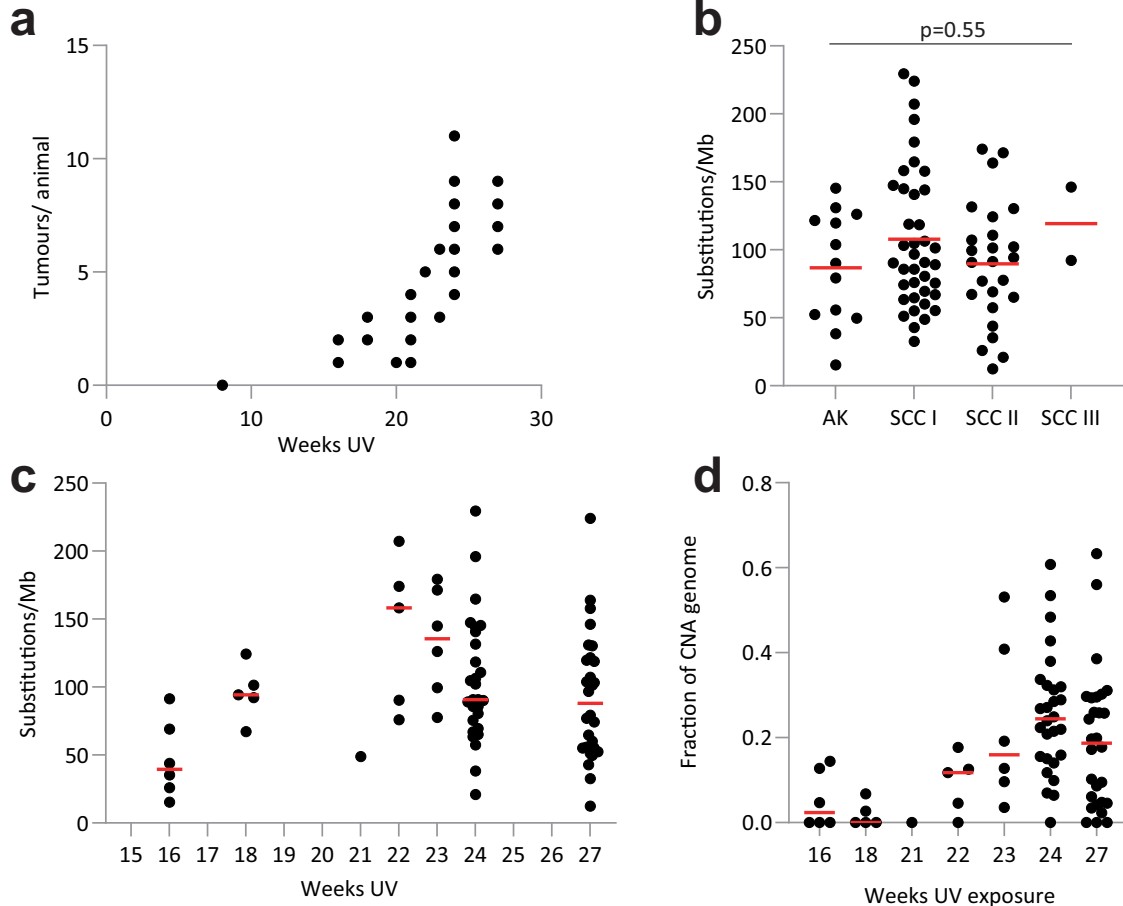

**Fig. 3 | Tumor genomes. a** Number of tumors per mouse at each time point (*n* mice 8 weeks, 10; 16 weeks, 4; 18 weeks 2; 20 weeks, 1; 21 weeks, 4; 22 weeks, 1; 23 weeks, 2; 24 weeks, 12; 27 weeks 4). Source data: Supplementary Data 7. **b** Mutational burden (single-base substitutions/Mb) across tumor histological grade estimated from multiregional targeted sequencing. Each point represents the average burden across the tumor. Number of tumors: AK *n* = 22, SCC grade I *n* = 87, SCC grade II *n* = 36, SCC grade III *n* = 2, *p* value from Kruskal–Wallis test. **c** Tumor mutational burden estimated from targeted sequencing (SBS/Mb). Each point represents the average burden in tumors at that time point. 16 wk, six tumors from four mice; 18 wk, five tumors from two mice; 21 wk, one tumor from one mouse; 22 wk, five tumors from one mouse; 23 wk, six tumors from one mouse; 24 wk, 28 tumors from four mice; 27 wk, 28 tumors from four mice. Source Data: Supplementary Data 8. **d** CNA over time. Each point represents the average copy number change across the tumor. Red lines indicate the median at each time point. Source Data: Supplementary Data 9.

cytological atypia of epidermal keratinocytes and involvement of adnexal structures according to ref. 81.

### Wholemount sample preparation

The whole back skin was cut into rectangular pieces of ~4 by 5 mm and incubated in PBS containing 5 mM EDTA at 37 °C for 2 h. The skin samples were transferred into PBS, and the epidermis was carefully scraped off using a curved scalpel by holding one corner of the sample with forceps. Samples were fixed in 4% paraformaldehyde in PBS for 30 min.

Epidermal samples were immunolabelled with p53 antibody (CM5, Vector Laboratories VP-P956) to highlight morphologically abnormal areas prior to dissection into 1–2mm² grids. Samples were blocked in staining buffer (0.5% Bovine Serum Albumin, 0.25% Fish Skin Gelatin, and 0.5% Triton X-100 in PBS) with 10% donkey serum for 1 h at room temperature. Wholemount samples were then incubated with p53 antibody (1:500) in staining buffer overnight, washed three times in PBS/0.2% Tween-20, incubated with fluorchrome-conjugated secondary antibodies (Invitrogen, A31572) for 2 h at room temperature and washed as before.

Epidermal wholemounts were dissected into 1–2 mm² grids and DNA extracted using QIAMP DNA microkit (Qiagen) by digesting overnight and following manufacturer's instructions. DNA was eluted using prewarmed AE buffer, where the first eluent was passed through the column two further

times. Flash-frozen liver DNA was used as the germline control, and DNA was extracted as for the epidermal samples.

### Tumors

The time of tumor appearance, tumor number and size were monitored three times per week. Euthanasia (using $CO_2$ gas in accordance with the requirements of the Veterinary Office of the Canton Vaud (Switzerland) and Federal Swiss Veterinary Office Guidelines) took place when the largest tumor volume reached 0.5 cm³ or in the case of 10% weight loss compared to the initial weight. These limits were not exceeded in any experiments. Tumors, non-tumoural dorsal skin and ventral skin samples were collected.

### Tumor grading

Tumors were graded as previously described[79]. Mouse skin tumors were paraformaldehyde-fixed and paraffin-embedded. Tissue sections (4 µm) were stained with hematoxylin/eosin. Histological analysis of actinic keratosis and tumor classification were performed in a blind manner by a dermatopathologist.

SCCs were classified according to the Broders' classification based on the degree of SCC keratinization and of keratinocyte differentiation[80]. This classification is as follows: SCC Grade I: 75% keratinocytes are well

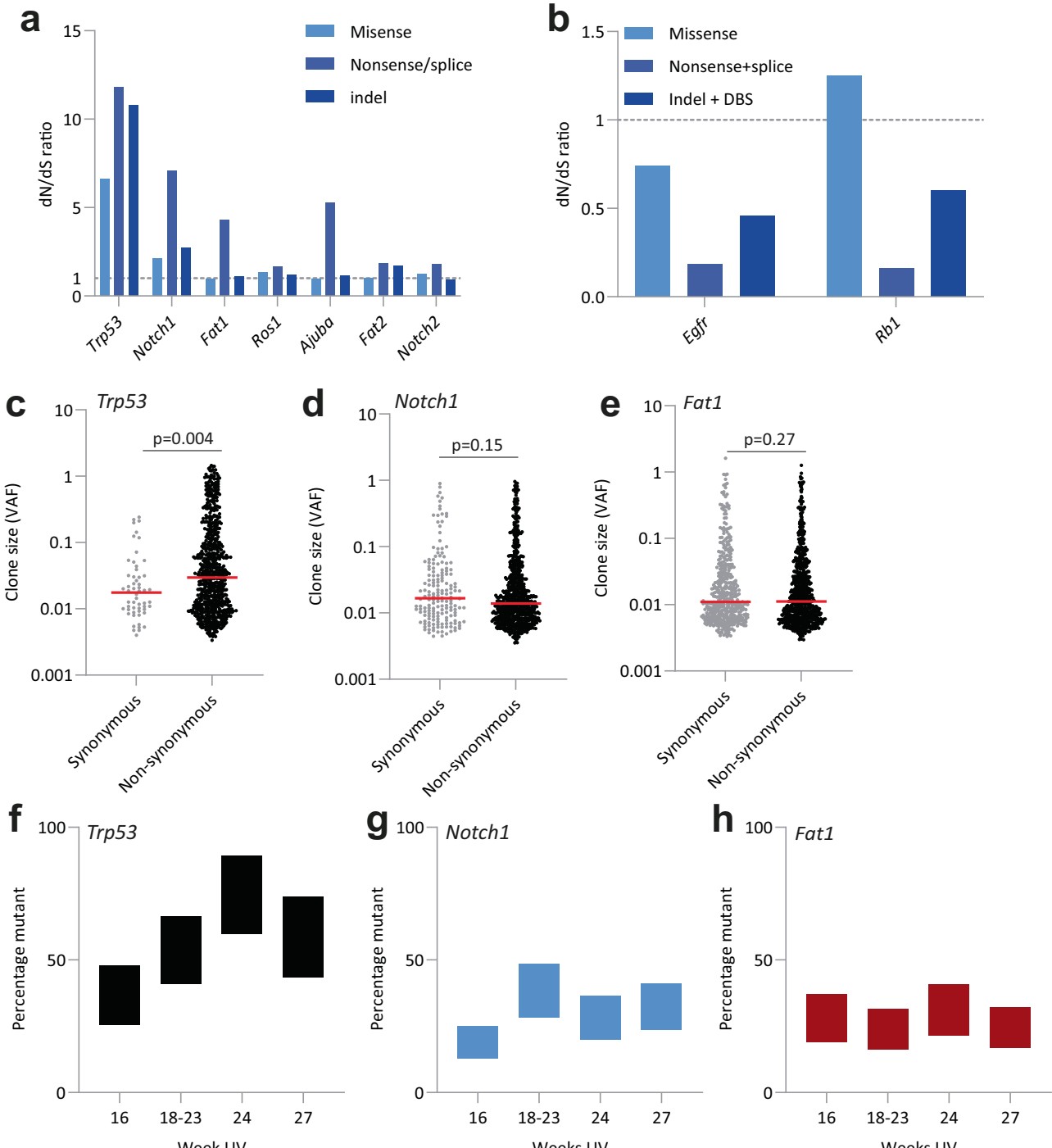

**Fig. 4 | Mutant selection in tumors. a** Genes under positive selection in tumors according to dN/dS. Mutations in all mice across all time points are pooled. $q < 0.01$, dNdS ratio >1.3 for nonsense/splice mutations. **b** Genes under negative selection in tumors according to dN/dS. Mutations in all mice across all time points are pooled $q < 0.01$, dNdS ratio <0.6 for nonsense/splice mutations. Source Data: Supplementary Data 12. **c–e** Distribution of synonymous and nonsynonymous clone size within tumors for clones mutant for *Trp53* (**c**), *Notch1* (**d**), and *Fat1* (**e**). Only tumors with three slices/ tumor are used ($n = 68$). Each dot represents a clone. *P* value determined by the Wilcoxon test. Source Data: Supplementary Data 13. **f–h** Percentage of tumor colonized by cells mutant for *Trp53* (**f**), *Notch1* (**g**), and *Fat1* (**h**), estimated from summed VAF at each time point. Data shows average across all tumors for each time point, 16 wk $n = 6$, 18-21 wk $n = 17$, 24 wk $n = 28$, 27 wk $n = 28$. Upper and lower bounds indicate uncertainty in copy number and multiplicity of mutations per cell. Source Data: Supplementary Data 14.

differentiated; SCC Grade II: >50% keratinocytes are well differentiated; SCC Grade III: >25% keratinocytes are well differentiated, and SCC Grade 4: <25% keratinocytes are well differentiated. Actinic keratoses were defined histologically and classified as grade I, II or III based on the degree of cytological atypia of epidermal keratinocytes and involvement of adnexal structures according to ref. 81.

**Wholemount sample preparation**

The whole back skin was cut into rectangular pieces of ~4 by 5 mm and incubated in PBS containing 5 mM EDTA at 37 °C for 2 h. The skin samples were transferred into PBS, and the epidermis was carefully scraped off using a curved scalpel by holding one corner of the sample with forceps. Samples were fixed in 4% paraformaldehyde in PBS for 30 min.

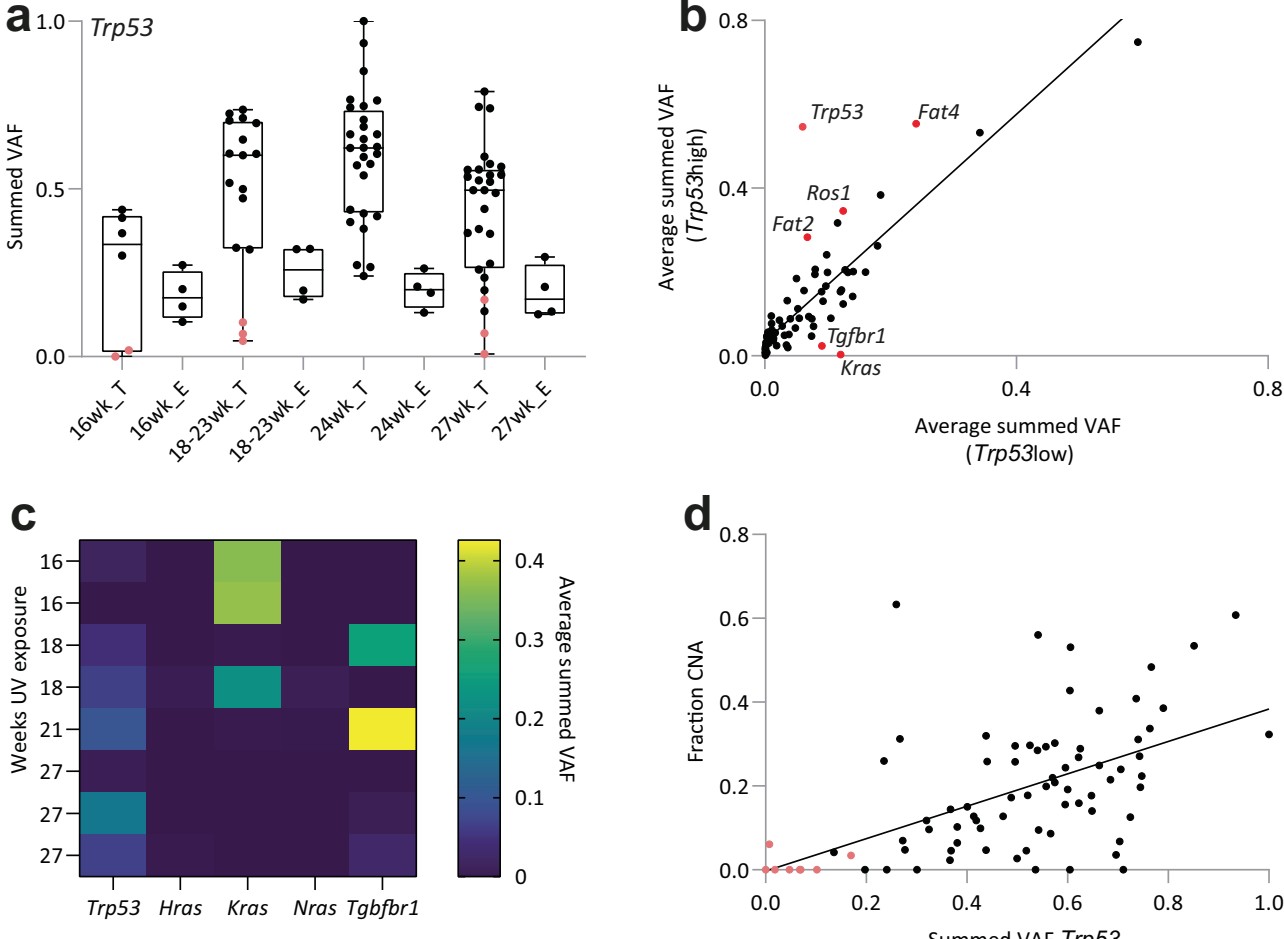

**Fig. 5 | Driver mutations in tumors. a** Fraction of tissue (summed VAF) with nonsynonymous mutants of *Trp53*. Dots are tumors (T) or epidermis (E). Tumors in pink have a lower proportion of mutant *Trp53* than the epidermis in the same animal. Central bar is median, box indicates quartiles, bars indicate range. Comparison of summed VAF in epidermis and tumor $p = 4.8 \times 10^{-16}$, two-tailed nested ANOVA. Source Data: Supplementary Data 15. **b** Selection in *Trp53*-high and -low tumors. Summed VAF (proportion of mutant tissue) is plotted for each sequenced gene; each dot is a gene, line indicates equivalence in both groups. two-tailed Z-test with Benjamini–Hochberg multiple test correction for outliers *Fat2* $p = 0.002$, *Fat4* $p = 0$, *Ros1* $p = 0.014$, *Kras* $p = 0$, *Tgfbr1* $p = 0.029$. Source Data: Supplementary Data 15. **c** Heatmap of tumors with fraction of mutant *Trp53* lower than the epidermis. The average summed VAF of other known tumor drivers is shown. Source Data: Supplementary Data 15. **d** Fraction of genome instability (fraction CNA) in relation to proportion of tumor tissue carrying *Trp53* mutations (summed VAF *Trp53*). Tumors identified as *Trp53* low (Fig. 5a) are marked as pink dots. $r^2 = 0.3$, $p < 0.0001$ linear regression. Source Data: Supplementary Data 11.

Epidermal samples were immunolabelled with p53 antibody (CM5, Vector Laboratories VP-P956) to highlight morphologically abnormal areas prior to dissection into 1–2 mm² grids. Samples were blocked in staining buffer (0.5% bovine serum albumin, 0.25% fish skin gelatin, and 0.5% Triton X-100 in PBS) with 10% donkey serum for 1 h at room temperature. Wholemount samples were then incubated with p53 antibody (1:500) in staining buffer overnight, washed three times in PBS/0.2% Tween-20, incubated with fluorchrome-conjugated secondary antibodies (Invitrogen, A31572) for 2 h at room temperature and washed as before.

Epidermal wholemounts were dissected into 1–2 mm² grids and DNA extracted using QIAMP DNA microkit (Qiagen) by digesting overnight and following manufacturer's instructions. DNA was eluted using prewarmed AE buffer, where the first eluent was passed through the column two further times. Flash-frozen liver DNA was used as the germline control, and DNA was extracted as for the epidermal samples.

**Tumor sample preparation**

Tumors were frozen in OCT and perpendicularly sectioned in a sequential manner. A thickness of 50 μm sections for sequencing were used. Tumors were stained, and tumor epithelia were collected for sequencing on the same day. Tumor sections were incubated in staining buffer (0.5% bovine serum

albumin, 0.25% fish skin gelatin, 0.5% Triton X-100, 0.1% saponin, and 1 mM EDTA in PBS) supplemented with 10% donkey serum for 15 min. Keratin 14 (Covance PRB-155P) antibody (used to label epithelia) was conjugated with Alexa Fluor 555 using an antibody labeling kit following the manufacturer's instructions (Thermo Fisher Scientific A20181). Sections were incubated with directly conjugated Keratin 14 antibody for 20 min in staining buffer, followed by two washes in 0.1% saponin PBS for 10 min. Positively stained Keratin 14 areas were dissected under fluorescent Leica M165 FC microscope and used for DNA extraction. Stroma was discarded. DNA was extracted as described above.

**Targeted sequencing**

DNA was sequenced at high depth using a custom bait panel (Agilent SureSelect) covering murine homologs of 74 genes implicated in human cancers, as described previously[82]. A list of genes covered by the bait set can be found in Supplementary Data 2.

About 200 ng of genomic DNA was fragmented (average size distribution ~150 bp, LE220, Covaris Inc), purified, libraries prepared (NEB-Next Ultra II DNA Library prep Kit, New England Biolabs), and index tags applied (https://emea.illumina.com/products/by-type/accessory-products/idt-pcr-index-sets.html). Index-tagged samples were amplified (six cycles of

**Fig. 6 | Genetic diversity in tumors. a, b** Fraction of tumor mutant for *Trp53* after 24 (**a** *p* = 0.51) and 27 (**b** *p* = 0.22) weeks of UV exposure. Each column is a different mouse (animals 1–4) and each point is a tumor. The medians from each animal are also plotted. Source Data: Supplementary Data 15. **c, d** Tumor mutational burden after 24 (**c** *p* = 0.3) and 27 (**d** *p* = 0.5) weeks of UV exposure. Each column is a different mouse (1–4) and each point is a tumor. The medians from each animal are also plotted. Source Data: Supplementary Data 10. **e, f** Fraction of genome which shows CNA at 24 (**e** *p* = 0.89) and 27 (**f** *p* = 0.68) weeks of UV exposure. Each column is a different mouse (1–4) and each point is a tumor. Red lines indicate medians from each mouse. *p* value calculated with ANOVA. Source Data: Supplementary Data 11. **g, h** Histology of tumors from each mouse at 24 (**g** *p* = 1) and 27 (**h** *p* = 0.99) weeks UV exposure. *p* value calculated with Kruskal–Wallis. *n* tumors sequenced (histology)/mouse 24 weeks, mouse 1,5(6); mouse 2,6(6); mouse 3,10(11); mouse 4,7(9) tumors 27 weeks, mouse 1, 8(8); mouse 2, 7(9); mouse 3, 6(6); mouse 4, 7(7). Source Data: Supplementary Data 9.

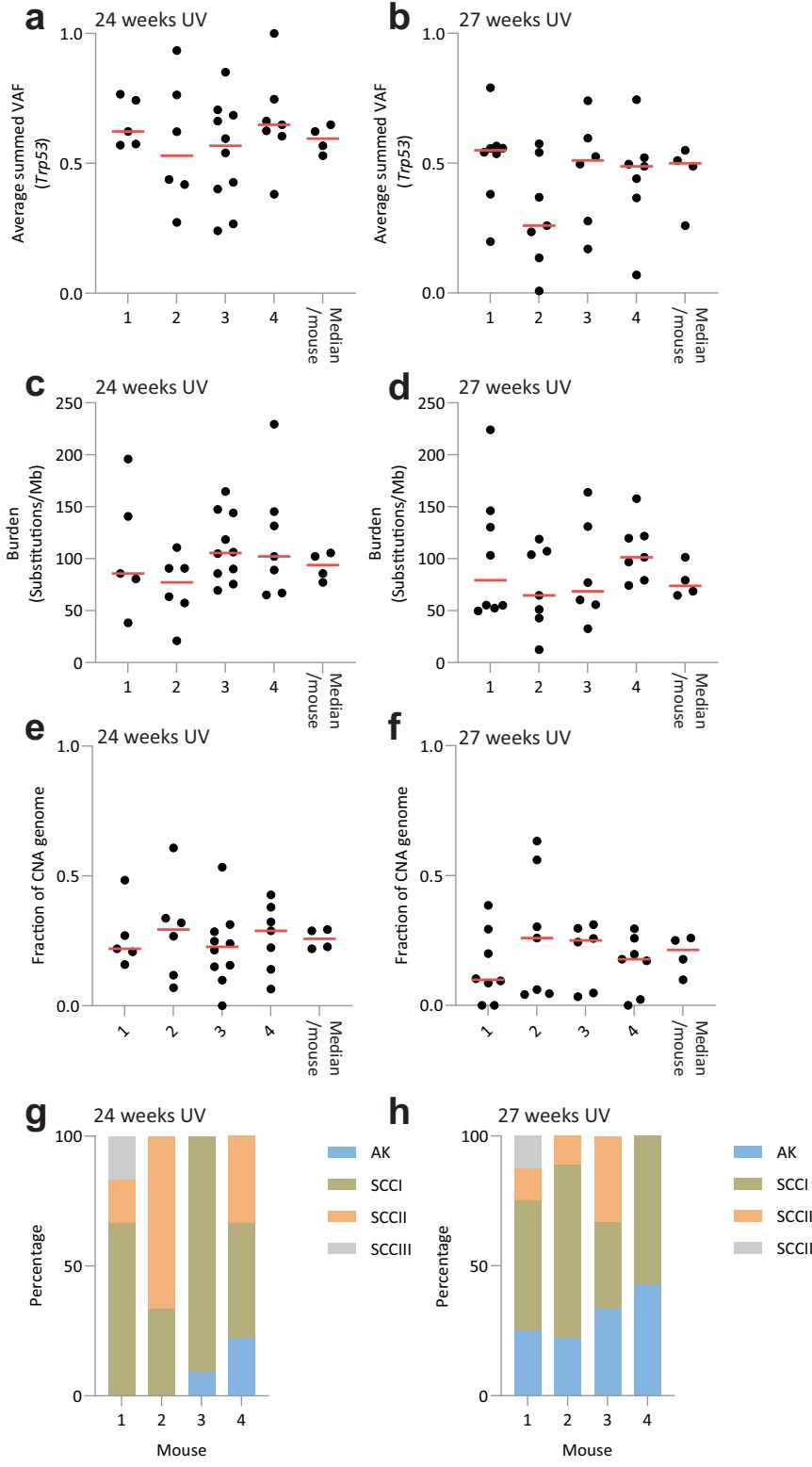

PCR, KAPA HiFi kit, KAPA Biosystems), quantified (Accuclear dsDNA Quantitation Solution, Biotium), and then pooled together in an equimolar fashion. About 500 ng of pooled material was taken forward for hybridization, capture and enrichment (SureSelect Target enrichment system, Agilent technologies), normalized to ~6 nM), and submitted to cluster formation for sequencing on HiSeq2500 (Illumina) to generate 75 bp paired-end reads.

BAM files were mapped to the GRCh37d5 reference genome using BWA-mem (version 0.7.17)[83]. Duplicate reads were marked using SAMtools (v1.11)[68]. Depth of coverage was also calculated using SAMTools to exclude reads which were: unmapped, not in the primary alignment, failing platform/vendor quality checks or were PCR/Optical duplicates. BEDTools (version 2.23.0) coverage program was then used to calculate the depth of coverage per base across samples[84] (Supplementary Figs. 1c, d—epidermis, 4e— tumor).

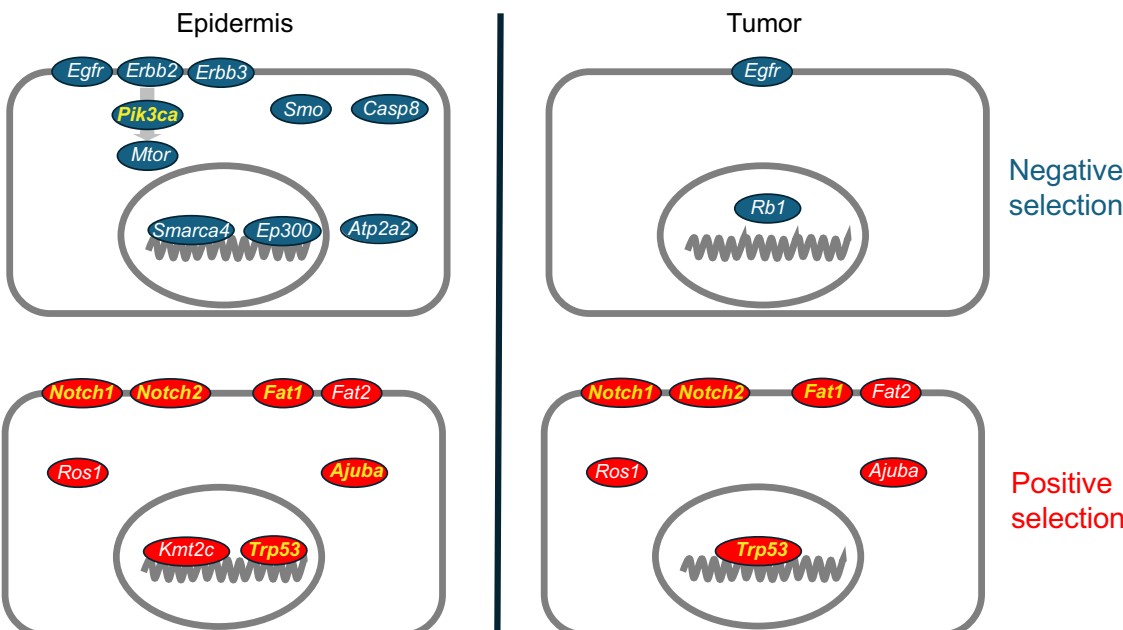

**Fig. 7 | Comparison of mutant gene selection measured by dN/dS in epidermis and tumors.** Negatively selected genes with dN/dS for nonsense mutations <0.7 and $q < 0.01$, blue, and positively selected mutants dN/dS for nonsense mutations >1.3 and $q < 0.01$, red, are shown. Genes in yellow are also selected in the human sun-exposed epidermis and AK/cSCC. Source Data: Supplementary Data 6, 12.

## Nanoseq

About 50 ng of DNA was used for DupSeq library preparation according to ref. 34. Briefly, DNA was digested with mung bean nuclease, A-tailed, repaired and tagged. 0.3 fmol of indexed tagged library were sequenced with 14 PCR cycles before quantifying and sequencing on Novaseq6000 (Illumina) with 150 bp PE reads to give a median 0.6x duplex coverage. About 3 fmol of indexed tagged libraries generated from DNA extracted from the liver of the same mouse was used as the germline control for calling of SNPs and indels. For both SNV and indels, only calls which passed all the defined filters were used https://github.com/cancerit/NanoSeq.

## Mutation calling

For targeted sequencing data, sub-clonal mutation variant calling was made using the deepSNV R package (also commonly referred to as ShearwaterML), version 1.21.3, available at https://github.com/gerstung-lab/deepSNV, used in conjunction with R version 3.3.0 (2016-05-03)[35]. Variants were annotated using VAGrENT[85].

Mutations called by ShearwaterML were filtered using the following criteria

- Positions of called SNVs must have a coverage of at least 100 reads
- Germline variants called from the same individual were removed from the list of called variants.
- Adjustment for FDR and mutations demanded support from at least one read from both strands for the mutations identified.
- Pairs of SNVs on adjacent nucleotides within the same sample are merged into a dinucleotide variant if at least 90% of the mapped DNA reads containing at least one of the SNV pair, contained both SNVs.
- Identical mutations found in multiple contiguous tissue biopsies are merged and considered as a single clone in order to prevent duplicate clone counting.

ShearwaterML was run with a normal panel of approximately 16k (tumor) or 31k (epidermis) reads.

## Calculations of the percentage mutant epithelia

The percentage of mutant tissue was estimated from the variant allele frequency, allowing for the uncertainty over copy number and nesting of mutants. dN/dS analysis was performed using https://github.com/im3sanger/dndscv[35].

## Copy number analysis

The copy number analysis was performed with the pipeline defined in Abby et al.[17], which was slightly modified in the following ways: (1) The segmentation gamma parameter was increased from 10 to 30. This parameter controls the likelihood that the segmentation starts a new segment. (2) Instead of the mean logR (log ratio, sample coverage normalized with coverage of the matched normal and corrected for GC content) per segment, the median logR was used as a summary statistic. This decreases the sensitivity of the gain/loss calling step to outlier data. We found these two changes reduce spurious segments, resulting in low confidence calls on these data. A bin size of 100 Kb was used with QDNAseq.

A parameter sweep was performed to investigate how segmentation behavior changes with different penalty (Segmentation γ) and minimum-size (k-min) settings. We compared five runs with different parameters (γ = 10, default γ = 30, k_min = 2, γ = 50, and γ = 100) and computed consistent run-level metrics: the number of segments per sample, segment-length distributions (absolute and normalized), and copy number class composition by summed segment length. These metrics permit direct comparison of segmentation granularity, the relative contribution of different length classes to total segmented bases, and whether copy-number balance (gain/loss/normal proportions) is preserved across parameter choices.

The vast majority of segments are concentrated in very large segments (> 50 Mb) for all runs (default γ = 30 → ~89.9%; γ = 10 → ~81.6%; γ = 100 → ~95.3%), indicating that we are conservatively calling chromosome-scale events that dominate the total segmented length (Supplementary Fig. 6a). This trend is as expected. Increasing γ leads to a higher segmentation penalty and fewer segments (Supplementary Fig. 6b). Similarly, decreasing the gamma leads to more segments, specifically, substantially more short segments (<1 Mb: 304 vs default = 169). However, copy-number class composition by summed length is highly stable across runs (for the default run, gain = 5.36%, loss = 5.66%, normal = 88.98%), with only small shifts for other γ values. (Supplementary Fig. 6c).

The parameter sweep serves as an explicit robustness check, which indicates that chromosome-level CNA patterns are robust to reasonable segmentation-parameter choices for the sequencing data.

The code of this pipeline is available[86].

## Clonality analysis

A tumor was assigned the label monoclonal or multifocal. A tumor was assigned as monoclonal when it fit one of the following categories (in order of priority): (1) At least five SNVs and at least one CNA >10 Mb were present in all samples, (2) at least 20 SNVs were present in all samples, (3) at least five SNVs present in all samples excluding samples without detected SNVs or CNAs from the edge of the tumor (in this scenario a sample was taken from just beyond the tumor boundary and therefore was completely normal), and (4) at least five SNVs were present in all samples and the CNA profile did not contain any alterations >10 Mb.

A tumor was assigned as multifocal when it fit one of the following categories (in order of priority): (1) samples did not share any clonal SNVs nor CNAs, at least two samples had separate sets of at least five SNVs and separate CNAs >10 Mb, (2) at least two samples had separate sets of at least five SNVs and the CNA profiles did not contain any alterations >10 Mb.

The remaining tumors were manually inspected and consistently yielded separate clonal expansions consistent with a multifocal label supported by SNVs, but with few conflicting CNAs. There is emerging evidence of parallel evolution of CNAs during a tumor's lifetime[87]. The remaining set of tumors were therefore assigned to be multifocal due to the support from SNVs.

## Statistics and reproducibility

Animals were randomly assigned to treatments and time points. Numbers of mice at each time point are shown in Supplementary Table 1. Investigators were not blinded to experimental time points. Statistical tests, as described in figure legends, were two-tailed unless noted.

## Animal ethics

All experiments involving animals were approved by the Veterinary Office of the Canton Vaud, Switzerland (VD1528.5) in accordance with the Federal Swiss Veterinary Office Guidelines and conform to the Commission Directive 2010/63/EU. We have complied with all relevant ethical regulations for animal use.

## Data availability

The sequencing datasets generated during the study are available at the European Nucleotide Archive (ENA, https://www.ebi.ac.uk/ena/browser/home) with the following dataset accession numbers: ERP166413 (epidermal sequencing dorsal and ventral). ERP166414 (tumor sequencing). ERP141605 (nanoseq of dorsal epidermis). Numerical source data is provided as Supplementary Data 2–15.

## Code availability

The code used for copy number analysis is available on https://github.com/PHJonesGroup/Skrupskelyte_etal_SI_code and has been deposited on Zenodo, https://zenodo.org/records/17734456[86].

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

## Acknowledgements

We thank Maude Delacombaz for expert assistance with Histology and Lisa Cherbuin, Marie Sgandurra, Lionel Mury, Hélène Garcia, and Dr Christine Pich for assistance with UV exposure and sample harvesting. This work was supported by grants from the Wellcome Trust to the Wellcome Sanger Institute (098051, 296194, and 220540/Z/20/A) and Cancer Research UK Program Grants to P.H.J. (C609/A17257 and C609/A27326). B.H. is supported by a Royal Society URF grant (UF130039). L.M. is supported by grants from the Swiss National Science Foundation (31003A-169232), SKINTEGRITY.CH collaborative research and the Etat de Vaud, University of Lausanne.

## Author contributions

G.S., C.W., J.K., and E.G. performed experiments. J.C.F., S.D., N.B., R.S., J.F., T.Q., and C.K. analysed data. I.A. performed statistical analysis supervised by B.H. J.C.F. and P.H.J. wrote the manuscript with input from all other authors. L.M. and P.H.J. supervised the research.

## Competing interests

The authors declare no competing interests.
