## [Transparent Peer Review file · Communications Biology]

The dynamics of mutational selection in cutaneous squamous carcinogenesis

Corresponding Author: Professor Philip Jones

Version 0:

Reviewer comments:

Reviewer #1

(Remarks to the Author)

The authors present an informative study of carcinogenesis and cancer evolution in a model of UV induced squamous cutaneous carcinoma in mice, using highly accurate DNA sequencing and two criteria comparing synonymous and non-synonymous mutations as criteria for negative selection and sequencing both malignant and normal tissues. The animal model allow interrogation of many individual tumors.

They identify two major pathways of carcinogenesis in this condition, TP53 dependent and TP53 independent. Notably, a single mouse harbors tumors of both types, i.e. heterogeneity between animals does not contribute substantially to overall heterogeneity.

Genes are identified that appear to be selected normal tissue but are not drivers in the carcinomas. In addition, negatively selected genes in tissues are not always negatively selected in the carcinoma, perhaps due to Hill-Robertson effects. This work has novel methodology and interesting conclusions and deserves to be published in my opinion, after minor revision.

I have several comments concerning potential revisions:

1. The distinction between dN/dS and VAF_N/VAF_S should be spelled out in simple equations in the main body and described conceptually in clear fashion. This is a key point for understanding the paper, and not all readers may be familiar with it.
2. The distinction in #1 above also carries through to the distinction between single molecule and bulk sequencing. This is alluded in lines 118-122, but again it is not explained sufficiently to be clear to a broad audience.
3. In the discussion, it is stated that genes selected in normal tissue but not in cancer might be genes not associated with proliferation. In my opinion, the authors should also consider the converse, i.e. that genes may contribute to proliferation but not to cancer if the large areas of normal tissue or benign/premalignant lesions such as colonic polyps are converted inefficiently to carcinomas. Beckman (Seminars in Cancer Biology, 2010) discusses this point in colon cancer by comparing the incidence of cancer development in hereditary non-polyposis colon cancer (HNPCC) and familial adenomatous polyposis (FAP), where the incidence curves are similar in spite of many more polyps in the latter. This line of thinking may have implications in that models like the Fearon-Vogelstein pathway for colorectal cancer are derived from sequencing of premalignant lesions, but this analysis inherently assumes that carcinomas primarily arise from "premalignant" polyps. The authors may wish to comment on this general issue, as I believe their results shed some light on it.

Reviewer #2

(Remarks to the Author)

In this study, the authors used mouse model to investigate the development of cutaneous squamous carcinoma. Importantly, they sequenced both the normal and tumour tissues to study the selection pressure on genes that might drive the expansion of normal and tumour clones over time. The main findings are that most tumors are driven by Trp53 mutations, while a minority arise from activating Kras mutations. Interestingly, the Trp53 mutations and other genes which showed signals of positive selection are similar in normal tissues, suggesting that these are early driver events. However, the genes that showed negative selection are different between normal and tumour tissues. Overall, the study is well-designed, and the results strongly support the authors' conclusions.

Minor comments for improvement:

- a) Are the Trp53 mutations observed in normal and tumour tissues clustered within specific domains (e.g., DNA-binding sites), or are they distributed throughout the gene?
- b) In Fig. 2g–h, the association of positively selected genes with area is unclear—are these observations limited to tumors with Trp53 mutations only, or do they also include co-occurrence with other positively selected genes (e.g., Notch1, Fat1)? An oncoplot/heatmap would help visualize the frequency of such co-occurrences, if any.
- c) In Fig. 5, it is interesting that some tumors lacking Trp53 mutations instead harbor Tgfbr1 or Kras alterations, suggesting an alternative route. However, this accounts for only 5 of 8 tumors. What about the remaining cases?
- d) In Fig. 5d, the tumours with increasing Trp53 VAF have been shown to positively correlate with the CNA fraction. Whether any of the CNA observed in these tumour affect Trp53, leading to biallelic inactivation?

Reviewer #3

(Remarks to the Author)

Skrupskelyte et al. present a study using a mouse-model, longitudinal, normal to tumor analysis under controlled UV exposure, combining dN/dS with a clone size selection proxy (VAF_NS/VAF_S). They explicitly contrast positive and negative selection between normal epidermis and multiple synchronous tumors, proposing a two route cSCC model (major Trp53 driven, minority Ras/Tgfbr1-type drivers) and report attenuation of many normal skin "drivers" inside tumors, as well as negative selection on a dozen of genes.

Major:

1- Negative selection signatures on oncogenes were reported previously, eg recently shown in DOI:10.1038/s41467-024-50552-1, and indeed here they report negative selection signatures in Egfr, Erbb2, Pik3ca etc. The negative selection is reduced in tumors compared to normal cells, plausibly due to hitchhiking along with positively selected mutations. I do have some worries about these results – first is observing negative selection in tumor suppressors. For the Rb1 they muster some sort of an explanation (not sure if compelling but it's something) however there is also Smarca4 which they don't comment on. A second concern is that the list of negatively selected genes does not seem to contain the universally-essential genes (not drivers) that were reported in previous studies like spliceosome, RNA pol 2 or such. Hard to see how Egfr would be more essential than this.

The worry is about control for oligonucleotide and strand biases and hotspots (CTCF/cohesin; ETS binding sites) in the dN/dS, which are rather extreme in this case of practically exclusively UV signature cancers. This needs extra support. One example test would be to compare the low-impact missense to high-impact missense mutations e.g. CADD or whatever is available for mouse. Other tests (excluding hotspots, explicit control for pentanucleotide etc) could be done to ensure dNdScv is really doing what it should be. Qq plots should be checked. If they use covariate mode from dNdScv (i.e. not dNdScv_loc) I am not too convinced that a covariate model could be reliably fit with <100 loci sequenced.

2- The specific claims "Trp53 drives ~90% of tumors" and "two routes to cSCC" have moderate support; this could be strengthened.

First of these claims is inferred from ~10% tumors classed as Trp53-low, but "drive" implies causality; while analysis does show good VAFNS/VAFS >1 support, whether the precise 90% figure follows from this is not clear. Is it possible to get an estimate if there is a notable percentage of these are passenger Trp53 mutations? A priori, this would not be unexpected given large mutation burdens.

Second is built on 8 Trp-low tumors, outlier tests for Kras/Tgfbr1, and a modest CNA correlation. It is intriguing but perhaps underpowered to be definitive. It would be strengthened with eg. mixed effects model, including fixed effects (time, grade) and random effect for mouse, and then FDR-correcting gene-wise comparisons between Trp53-high vs low; this would address concerns about pseudoreplication.

3- The CNA result, that they increase with UV duration and seem to show recurrent chromosomal biases (perhaps selected, although hard to tell) is interesting. I do have some questions about inferring CNA from panel sequencing with a small panel such as here. Ideally we would have validation from (at least shallow) WGS, but it is as it is, I do not expect them to add this for this study, but having extra clarification and perhaps supporting analysis would help. I understand that the CNA calling mostly draws on off-target reads (fine, there should be plenty of those) however these will have a certain baseline coverage distribution determined by sequence similarities. This does not seem like it will be sufficiently modelled by the G+C content. Is the matched-normal comparison enough to annul these biases? (which would register as recurrent CNA segments). How is the change in the segmentation coarseness parameter justified? Would the specific CNA calls remain robust after some perturbation of the data (e.g. coverage reduction, sample removal or such)?

Minor:

"BAM files were mapped to the GRCh37d5 reference genome using BWA-mem" this is a human reference genome, not mouse. Copy-paste error? Specify the actual reference used. Also I did not notice mention of the transcript set that was used.

Typos: abstract "carinogenesis", spacing in "candidatesfor".

Version 1:

Reviewer comments:

Reviewer #1

(Remarks to the Author)

The authors have addressed my comments very well. I recommend acceptance

Reviewer #2

(Remarks to the Author)

The authors have addressed all my comments satisfactorily. I have no further comments. Congratulations to the team for this excellent work.

Reviewer #3

(Remarks to the Author)

The authors have addressed my queries satisfactorily by various clarifications and some additional analyses. I would be supportive about seeing the study published in the current form.

Response to reviewers COMMSBIO-25-7321-T

We are most grateful to the reviewers for their insightful comments which have significantly improved the revised manuscript. In the below, our comments are in blue.

Reviewer #1 (Remarks to the Author):

The authors present an informative study of carcinogenesis and cancer evolution in a model of UV induced squamous cutaneous carcinoma in mice, using highly accurate DNA sequencing and two criteria comparing synonymous and non-synonymous mutations as criteria for negative selection and sequencing both malignant and normal tissues. The animal model allow interrogation of many individual tumors.

They identify two major pathways of carcinogenesis in this condition, TP53 dependent and TP53 independent. Notably, a single mouse harbors tumors of both types, i.e. heterogeneity between animals does not contribute substantially to overall heterogeneity.

Genes are identified that appear to be selected normal tissue but are not drivers in the carcinomas. In addition, negatively selected genes in tissues are not always negatively selected in the carcinoma, perhaps due to Hill-Robertson effects.

This work has novel methodology and interesting conclusions and deserves to be published in my opinion, after minor revision.

We are most grateful to the reviewer for their positive assessment and helpful critique.

I have several comments concerning potential revisions:

1. The distinction between dN/dS and VAF_N/VAF_S should be spelled out in simple equations in the main body and described conceptually in clear fashion. This is a key point for understanding the paper, and not all readers may be familiar with it.

Explaining dN/dS and VAF_N/VAF_S is indeed crucial the paper. We now include additional explanation, with simple, 'spelt out' equations of these metrics for biological readers. The text now reads:

'We applied two metrics of selection to normal and tumor tissue. The first was the widely used approach of estimating the ratio of the number of protein altering (missense, nonsense, essential splice, and insertions/deletions) to synonymous mutations, dN/dS, within each gene, so that for missense mutations for example, $dN/dS_{mis} = n_{missense}/n_{synonymous}$

where n is the total number of mutations of each type across the gene. The implementation of dN/dS used, dNdScv, controls for trinucleotide mutational signatures, sequence composition, and variation in mutation rates between genes, to avoid the known biases in dN/dS estimation^{1,2}. A normalised dN/dS value of 1 is neutral, values significantly above 1 after multiple test correction indicate positive selection, while values below 1 indicate negative selection. In this experiment, dN/dS

values reflect the likelihood that a nonsynonymous mutant clone will reach a detectable clone size compared with a synonymous mutant clone in the same gene.

A second measure of selection that may be suitably applied to the current analysis of normal tissue and tumors from a single cell lineage is the ratio of variant allele frequencies (VAF, defined as the proportion of sequencing reads reporting a mutation within each sample) of nonsynonymous and synonymous mutants in each gene (VAF_{ns}/VAF_s).

$VAF_{ns}/VAF_S = \text{sum of VAF of all non synonymous mutants} / \text{sum of all synonymous mutants in each gene.}$

Unlike dN/dS , the VAF of all types of non synonymous mutations is pooled in the calculation of VAF_{ns} . If a mutant gene drives clonal expansion, the VAF of nonsynonymous mutants will be significantly larger than the VAF of synonymous mutants in the same gene due to an increased proportion of nonsynonymous mutant cells in the sample giving a VAF_{ns}/VAF_S ratio significantly above 1. In contrast for neutral 'passenger' mutations, the VAF_{ns}/VAF_s ratio will be close to 1.' Lines 81-107.

2. The distinction in #1 above also carries through to the distinction between single molecule and bulk sequencing. This is alluded in lines 118-122, but again it is not explained sufficiently to be clear to a broad audience.

Again, this is a useful point. We now include a non-technical description of the key differences between these methods to guide the general biological reader as follows:

'Two methods of sequencing were employed. To detect mutant clones we used targeted sequencing of DNA extracted from small samples of normal tissue, where the presence of mutant clones is evidenced by mutant reads for each sequenced gene. As mutant clones may only be present in a small fraction of each tissue sample, we used the ShearwaterML algorithm, which builds a model of background sequencing errors at each base to identify variants with a statistical excess of mutation reads. This allowed detection of mutations clones in as few as 1% of the cells in a sample. To measure the mutational burden, the number of mutations across the genome, we used Nanoseq. This is a modification of duplex sequencing which allows the detection of somatic mutations across one third of the genome with an error rate $< 5 / 10^9$ base pairs³. Nanoseq can detect a mutation even if it occurs in just a single DNA molecule in a sample. Importantly for this experiment Nanoseq estimates of mutational burden are not affected by the expansion of mutant clones.' Lines 111-123.

3. In the discussion, it is stated that genes selected in normal tissue but not in cancer might be genes not associated with proliferation. In my opinion, the authors should also consider the converse, i.e. that genes may contribute to proliferation but not to cancer if the large areas of normal tissue or benign/premalignant lesions such as colonic polyps are converted inefficiently to carcinomas. Beckman (Seminars in Cancer Biology, 2010) discusses this point in colon cancer by comparing the

incidence of cancer development in hereditary non-polyposis colon cancer (HNPCC) and familial adenomatous polyposis (FAP), where the incidence curves are similar in spite of many more polyps in the latter. This line of thinking may have implications in that models like the Fearon-Vogelstein pathway for colorectal cancer are derived from sequencing of premalignant lesions, but this analysis inherently assumes that carcinomas primarily arise from “pre-malignant” polyps. The authors may wish to comment on this general issue, as I believe their results shed some light on it.

This is another helpful insight. We agree with the proposition that mutants that drive clonal expansions in normal tissue may be neutral in malignant transformation or even reduce cancer risk. The difference in cancer risk between HNPCC and FAP is an interesting example⁴. There is also work in the squamous esophagus, where the fittest mutant gene *NOTCH1*, is less prevalent in squamous carcinoma than in normal tissue, suggesting mutation may decrease the risk of malignant transformation^{5, 6}. In transgenic mouse models, loss of *Notch1* slows tumor growth by impairing oncogenic signalling⁵. In human sun exposed skin, however, the prevalence of *NOTCH1* and *FAT1* mutations in normal tissue is similar to that in keratinocyte cancers, suggesting these mutants may not contribute to cancer formation⁷.

We have revised the discussion as follows:

‘Mutants that drive clonal expansions in normal tissues or premalignant lesions may be neutral or even inhibitory in malignant transformation^{4, 5, 6, 7, 8}.’ Lines 423-425

Reviewer #2 (Remarks to the Author):

In this study, the authors used mouse model to investigate the development of cutaneous squamous carcinoma. Importantly, they sequenced both the normal and tumour tissues to study the selection pressure on genes that might drive the expansion of normal and tumour clones over time. The main findings are that most tumors are driven by Trp53 mutations, while a minority arise from activating Kras mutations. Interestingly, the Trp53 mutations and other genes which showed signals of positive selection are similar in normal tissues, suggesting that these are early driver events. However, the genes that showed negative selection are different between normal and tumour tissues. Overall, the study is well-designed, and the results strongly support the authors’ conclusions.

We are most grateful to the reviewer for their comments and suggestions.

Minor comments for improvement:

a) Are the Trp53 mutations observed in normal and tumour tissues clustered within specific domains (e.g., DNA-binding sites), or are they distributed throughout the gene?

This is an interesting point. We now include a supplementary figure showing the distribution of *Trp53* mutants in both epidermis and tumors:

Reviewer Figure 1 (Supplementary Fig. 3): Distribution of *Trp53* missense mutations in epidermis (upper panel) and tumors (lower panel).

We observed most mutations occur within the DNA binding domain. This is also the case with *TP53* missense mutants in human sun exposed skin and cutaneous squamous cell carcinoma^{9, 10}. Mutational hotspots at R210, R270 and R275 correspond to the TP53 mutations R213, R270 and R278 which are common in human cancer (<https://cancer.sanger.ac.uk/cosmic/gene/analysis?ln=TP53>). Transgenic mouse studies show that the R270H mutants are tumorigenic in mice, though mouse models do not exist for R275 and R210 mutants¹¹.

We describe these results as follows:

‘Most *Trp53* missense mutants resulted in codon changes within the DNA binding domain, in both epidermis and tumors (**Supplementary Fig. 7**). A similar distribution of TP53 missense mutants is seen in human sun exposed skin and cutaneous squamous cell carcinoma^{2, 7, 9, 10}. Lines 277-280.

b) In **Fig. 2g–h**, the association of positively selected genes with area is unclear—are these observations limited to tumors with *Trp53* mutations only, or do they also include co-occurrence with other positively selected genes (e.g., *Notch1*, *Fat1*)? An oncoplot/heatmap would help visualize the frequency of such co-occurrences, if any.

Thank you. All tumors and all mutant genes were included in the analysis of selection in tumors, now **Fig. 5 a, b**. We also include heat maps showing of positively selected

mutants in tumors in **Supplementary Fig. 8**. We have now examined selection in *Notch1* and *Fat1* high vs low VAF tumors finding no significant differences in selection other than in *Notch1* and *Fat1* themselves (**Reviewer Fig. 2, Supplementary Fig. 9**).

Reviewer Figure 2 (Supplementary Fig. 9) a, b Fraction of tissue (summed VAF) with nonsynonymous mutants of *Notch1*, **a**, and *Fat1*, **b**. Dots are tumors (T) or epidermis (E), Tumors in orange have a lower proportion of mutant *Notch1* or *Fat1* than epidermis in the same animal (**Supplementary Table 13**). Central bar is median, box indicates quartiles, bars indicate range. Comparison of summed VAF in epidermis and tumour $p=4.8 \times 10^{-16}$, 2 tailed nested Anova. **c,d** Selection in mutant-high and-low tumors, **c**, *Notch1*, **d**, *Fat1*. Summed VAF (proportion of mutant tissue) is plotted for each sequenced gene, each dot is a gene, line indicates equivalence in both groups. 2 tailed Z-test with Benjamini-Hochberg multiple test correction for outliers. Orange, significantly different mutant genes, *Notch1* $p= 1.2 \times 10^{-11}$, *Fat1* $p= 9.0 \times 10^{-11}$.

We cite this figure in the text as follows:

‘...there were no differences in mutant gene selection comparing tumors with a proportion of *Notch1* or *Fat1* mutant cells above or below that in the epidermis (**Supplementary Fig. 9**).’ Lines 297-299.

c) In Fig. 5, it is interesting that some tumors lacking Trp53 mutations instead harbor Tgfbr1 or Kras alterations, suggesting an alternative route. However, this accounts for only 5 of 8 tumors. What about the remaining cases?

Of the 3 remaining *Trp53* mutant low tumors, one (MD5435_B1) has a clonal mutation in *Notch2* and another *Ros1* and *Kmt2c* mutations (MD5438_B2). *NOTCH2* and *KMT2C* are recurrently mutated in human cutaneous squamous cell carcinoma^{12, 13, 14} as are *Kmt2c* and *Notch2* mutants in UV induced skin tumors in *Xpa* mutant mice¹⁵. *Kmt2c* was also identified as a tumor driver in a keratinocyte cancer transposon mutagenesis screen in mice¹⁶. Functional studies in human cells support a role for *KMT2C* in squamous transformation¹⁶. Conversely mutations in *ROS1* are rare in human skin squamous carcinoma and have not been shown to have a role in skin carcinogenesis in mice. MD5437_B1 carries no mutations in the sequenced genes. We presume that this tumor carries other genomic changes not detected by the targeted sequencing approach we used.

We now discuss this in the text as follows:

'Three *Trp53*low tumors harbored gain of function mutant *Kras* (one each of lesions with *Kras*^{G12D}, *Kras*^{G12C} and *Kras*^{P34L}) and two were mutant for *Tgfbr1* (one with clonal nonsense and missense mutations and the other with a missense mutation). (**Fig. 5b, c, Supplementary Fig. 5**). Functional studies in mice argue gain of function *Kras* and *Tgfbr1* mutations promote squamous skin tumor formation^{17, 18, 19, 20}. *Notch2* and *Kmt2c* mutations, found recurrently in mouse and human skin SCC, were present in two tumors^{12, 13, 14, 15}. In one tumor there were no detected mutations in the panel of sequenced genes.' Lines 301-307.

d) In Fig. 5d, the tumours with increasing *Trp53* VAF have been shown to positively correlate with the CNA fraction. Whether any of the CNA observed in these tumour affect *Trp53*, leading to biallelic inactivation?

This is an interesting point. *Trp53* mutation was associated with copy number loss at the *Trp53* locus in two tumors. However, due to the limitations of the off target read approach we used to estimate CNA, which limits our resolution to chromosome level events, this may be an underestimate. We now mention this in a paragraph on the limitations of the study in the discussion:

'Limitations of the study follow from the targeted sequencing approach used..... the resolution of the approach used to identify CNA is low, and changes smaller than a chromosome arm amplification or deletion may not be detected. The estimate of the proportion of the genome with CNA should be viewed as a lower bound. .' Lines 457-461.

Reviewer #3 (Remarks to the Author):

Skrupskelyte et al. present a study using a mouse-model, longitudinal, normal to tumor analysis under controlled UV exposure, combining dN/dS with a clone size selection proxy (VAF_NS/VAF_S). They explicitly contrast positive and negative selection between normal epidermis and multiple synchronous tumors, proposing a two route cSCC model (major *Trp53* driven, minority *Ras/Tgfbr1*-type drivers) and report attenuation of many normal skin "drivers" inside tumors, as well as negative selection on a dozen of genes.

Major:

1- Negative selection signatures on oncogenes were reported previously, eg recently shown in DOI:10.1038/s41467-024-50552-1 , and indeed here they report negative selection signatures in *Egfr*, *ErbB2*, *Pik3ca* etc. The negative selection is reduced in tumors compared to normal cells, plausibly due to hitchhiking along with positively selected mutations.

Thank you for highlighting this work, which we now cite in our discussion of negative selection in the introduction.

‘As well as positively selected mutants, identifying mutant genes that are negatively selected, particularly those negatively selected in tumors, may reveal therapeutic targets²¹. The existence of negative selection in human tumors was initially controversial, but more recent evidence argues it is widespread^{1, 21, 22, 23}. Many studies of normal tissues have too few synonymous mutations to identify negatively selected genes¹. However, studies with sufficient power have detected negative selection of somatic mutants in normal tissues, in the skin and oral epithelium^{7, 24}.’ Lines 60-66.

I do have some worries about these results – first is observing negative selection in tumor suppressors. For the *Rb1* they muster some sort of an explanation (not sure if compelling but it’s something) however there is also *Smarca4* which they don’t comment on.

The reviewer raises an important point. The negative selection of the tumor suppressor *Rb1* in tumors (*Rb1* mutants are neutral in the epidermis) is indeed surprising. However, experimental studies in mouse skin carcinogenesis are consistent with this observation. Briefly, conditional deletion of *Rb1* in *Rb1^{flox/flox}* adult mouse epidermis does not result in spontaneous tumor formation^{25, 26}. In chemical carcinogenesis of the skin, however, the size of tumors in *Rb1^{-/-}* epidermis is reduced due to tumor cell apoptosis compared with *Rb1* wild type controls, consistent with negative selection of *Rb1* in tumors we observe²⁶.

We now explain these results follows:

‘The negative selection of nonsense and essential splice *Rb1* mutants in tumors seems at odds with the tumor suppressor role of RB1 in a range of human cancers²⁷. However, this finding is consistent with the results of conditional *Rb1* deletion in the epidermis of adult *Rb1^{flox/flox}* mice²⁶. In a chemical carcinogenesis protocol, *Rb1* loss resulted in fewer and smaller tumors compared with wild type controls²⁶. The reduced size of the *Rb1^{-/-}* tumors is due to increased apoptosis in tumor cells, consistent with negative selection of *Rb1* in mouse skin tumors²⁶.’ Lines 444-450.

Smarca4 is negatively selected in the epidermis but not in tumors. Deletion of *Smarca4* in mouse epidermis results in a severe skin phenotype with a lethal loss of the epidermal barrier²⁸. *Smarca4* has an essential role in regulating expression of epidermal differentiation complex genes which is critical for differentiation²⁹.

We have changed the text as follows:

'Mutations disrupting the epigenetic regulators *Ep300* and *Smarca4*, the latter essential for epidermal differentiation....^{28, 29} Lines 195-196.

A second concern is that the list of negatively selected genes does not seem to contain the universally-essential genes (not drivers) that were reported in previous studies like spliceosome, RNA pol 2 or such. Hard to see how *Egfr* would be more essential than this.

'To be affordable at the high coverage required to identify rare variants in normal tissue we used a panel of genes implicated in squamous carcinogenesis. Unfortunately, the gene panel did not include universally essential genes. We now include a paragraph on the limitations of the study in the discussion that highlights this point.

'Limitations of the study follow from the targeted sequencing approach used. There may be additional positively selected mutants in genes not in the sequenced panel, and universal essential and other genes under negative selection were not included.' Lines 457-461.

The worry is about control for oligonucleotide and strand biases and hotspots (CTCF/cohesin; ETS binding sites) in the dN/dS, which are rather extreme in this case of practically exclusively UV signature cancers. This needs extra support. One example test would be to compare the low-impact missense to high-impact missense mutations e.g. CADD or whatever is available for mouse.

Other tests (excluding hotspots, explicit control for pentanucleotide etc) could be done to ensure dNdScv is really doing what it should be. Qq plots should be checked. If they use covariate mode from dNdScv (i.e. not dNdScv_loc) I am not too convinced that a covariate model could be reliably fit with <100 loci sequenced.

The reviewer highlights potential concerns about the limitations of dN/dS analysis with targeted sequencing and of dNdScv in particular. Working in mouse, we are limited in the driver discovery tools we can use, e.g. OncoDriveFML and CADD are human specific. We were drawn to dNdScv as this has been applied to deep targeted sequencing of 74-121 gene panels in sun exposed human skin with a strong UV signature, paralleling the mouse experiments reported here ^{2, 7, 10}.

One issue in applying dN/dS to small numbers of genes in targeted sequencing studies is that the mutational spectrum may not be reliable given the small number of loci sequenced and the predominant contribution of a few strongly selected mutant genes. To address this, we compared the mutational spectrum generated by targeted sequencing of epidermis with that from Nanoseq, which samples ca. 30% of the genome (**Reviewer Fig. 2 a,b**). The numbers of mutations at the 96 trinucleotide contexts were well correlated given the differences between a small sample of genes versus intergenic DNA (**Reviewer Fig. 2c**).

Reviewer Fig. 3 (Supplementary Fig. 2). Mutational spectra of targeted and nanoseq epidermal sequencing.

a, b. Mutational spectra of targeted sequencing (**a**) and Nanoseq (**b**). **c,** Correlation of spectra shown in **a** and **b**, each dot represents a trinucleotide context, Pearson's test.

We now include these results as a new **Supplementary Figure 2** which we cite in the text as follows:

'The mutational spectrum was dominated by C>T transitions, consistent with UV exposure, in both targeted and Nanoseq sequencing of the epidermis (**Supplementary Fig. 2**)².' Lines 163-164

To address the issue of covariates, we ran both dNdScv and NdScv_loc with and without covariates. Minor differences in dN/dS values were seen but the lists of selected genes did not change (Reviewer Fig. 4 and Reviewer tables 1 and 2).

Reviewer Figure 4. Comparison of global q values generated by dNdSloc and dNdScv in normal epidermis (a) and tumors (b) run with and without covariates (no_cv). Mutant genes under selection ($q < 0.01$) remain the same.

gene_name	qglobal_cv_normal	qglobal_cv_normal_no_cv	qall_loc_normal	qall_loc_normal_no_cv
Trp53	0	0	0	0
Notch1	0	0	0	0
Lrp1b	0	0	0	0
Fat3	0	0	0	0
Fat1	0	0	0	0
Fat4	2.97E-08	2.97E-08	6.61E-09	6.61E-09
Notch2	2.97E-08	2.97E-08	2.91E-09	2.91E-09
Erbb2	2.64E-07	2.64E-07	1.80E-08	1.80E-08
Kdr	3.89E-06	3.89E-06	2.24E-06	2.24E-06
Kmt2c	6.15E-06	6.15E-06	8.20E-07	8.20E-07
Ros1	1.04E-05	1.04E-05	2.94E-06	2.94E-06
Ajuba	1.76E-05	1.76E-05	2.94E-06	2.94E-06
Egfr	3.65E-05	3.65E-05	2.96E-06	2.96E-06
Ep300	8.62E-05	8.62E-05	9.52E-06	9.52E-06
Smarca4	0.000151465	0.000151465	1.79E-05	1.79E-05
Card11	0.000183434	0.000183434	9.63E-05	9.63E-05
Arid2	0.00206776	0.00206776	0.000236732	0.000236732
Fgfr3	0.00206776	0.00206776	0.000302505	0.000302505
Fat2	0.00206776	0.00206776	0.000547458	0.000547458
Smo	0.00206776	0.00206776	0.000302505	0.000302505
Erbb3	0.004256422	0.004256422	0.000663046	0.000663046
Atp2a2	0.005431068	0.005431068	0.000750801	0.000750801

Reviewer Table 1: Global q values from epidermis analysed with dNdScv and dNdSloc with and without covariates (no_cv).

gene_name	qglobal_cv_tumour	qglobal_cv_tumour_no_cv	qall_loc_tumour	qall_loc_tumour_no_cv
Trp53	0	0	0	0
Notch1	0	0	0	0
Lrp1b	4.12E-11	4.12E-11	3.51E-11	3.51E-11
Fat3	2.48E-05	2.48E-05	1.38E-05	1.38E-05
Fat1	0	0	0	0
Fat4	7.65E-06	7.65E-06	2.83E-06	2.83E-06
Notch2	0.002461614	0.002461614	0.000481652	0.000481652
Erb2	0.032617197	0.032617197	0.007206583	0.007206583
Kdr	0.002485055	0.002485055	0.001706441	0.001706441
Kmt2c	0.032745218	0.032745218	0.009488254	0.009488254
Ros1	9.08E-07	9.08E-07	2.68E-07	2.68E-07
Ajuba	1.44E-06	1.44E-06	2.30E-07	2.30E-07
Egfr	0.001149588	0.001149588	0.000125759	0.000125759
Smarca4	0.032745218	0.032745218	0.009488254	0.009488254
Fgfr3	0.033065819	0.033065819	0.009573084	0.009573084
Fat2	4.25E-06	4.25E-06	1.54E-06	1.54E-06
Smo	0.032617197	0.032617197	0.007206583	0.007206583
Erb3	0.032617197	0.032617197	0.008164656	0.008164656
Plk3ca	0.032617197	0.032617197	0.006447511	0.006447511
Fbxo21	0.032745218	0.032745218	0.007206583	0.007206583
Kmt2d	0.032617197	0.032617197	0.009488254	0.009488254
Rb1	0.004403297	0.004403297	0.000633423	0.000633423

Reviewer table 2: Global q values from tumors analysed with dNdScv and dNdSloc with and without covariants (no_cv).

We have not included these results in the paper at present but would be happy to do so if the editor or reviewers request it.

In terms of the functional impact of missense mutations, concerns have been raised over accuracy the simpler predictive tools available in mouse such as SIFT and PolyPhen2³⁰. More sophisticated modelling is required to predict the effects of missense mutations³¹. Saturation mutagenesis of the DNA binding domain of *TP53* has shown that the impact of mutants depends on the functional assay used, with 2D, 3D and in vivo models giving different results^{32, 33}. We submit that that accurately resolving which *Trp53* missense mutants have functional impact in squamous carcinogenesis is currently infeasible beyond the few already studied in vivo^{11, 33, 34}.

Missense mutant gene selection metrics may be confounded if positively and negatively selected missense mutants occur in in the same gene^{7, 21}. For this reason, we opted not to use missense dN/dS values to determine mutant gene selection. We now highlight that our criteria for a gene being under selection were a dN/dS of nonsense /essential splice mutants >1.3 for positive and <0.6 for negative selection (q<0.01).

We have amended the text to clarify this point:

‘As missense dN/dS ratios may be misleading if both gain and loss of function mutants occur in the same gene, we only considered the dN/dS for nonsense and essential splice mutant for determining which mutant genes were under selection⁷. After correction for mutational spectrum, genomic context and multiple testing, eight mutant genes, including *Trp53*, *Notch1*, *Notch2*, *Fat1*, and *Ajuba* emerged as being under

positive selection in the epidermis (dNdSnonsense/essential splice >1.3,q<0.01)' Lines 179-184.

2- The specific claims “Trp53 drives ~90% of tumors” and “two routes to cSCC” have moderate support; this could be strengthened.

First of these claims is inferred from ~10% tumors classed as Trp53-low, but “drive” implies causality; while analysis does show good VAFNS/VAFS >1 support, whether the precise 90% figure follows from this is not clear. Is it possible to get an estimate if there is a notable percentage of these are passenger Trp53 mutations? A priori, this would not be unexpected given large mutation burdens.

As discussed above, determining the proportion of missense *Trp53* mutations that are neutral in squamous carcinogenesis is infeasible for almost all mutants, so we cannot determine the proportion of passengers³⁰.

We have therefore modified the text, removing ‘drive’, as follows:

‘Nonsynonymous *Trp53* mutants were positively selected in both epidermis and tumors and present in 90% of tumors’ Lines 39-40.

Second is built on 8 Trp-low tumors, outlier tests for Kras/Tgfbr1, and a modest CNA correlation. It is intriguing but perhaps underpowered to be definitive. It would be strengthened with eg. mixed effects model, including fixed effects (time, grade) and random effect for mouse, and then FDR-correcting gene-wise comparisons between Trp53-high vs low; this would address concerns about pseudoreplication.

We have explored mixed effect models as suggested by the reviewer. A mixed effects model with fixed time and random effect for mouse shows a significant effect for tumor type (*Trp53*high, vs *Trp53*low), but no significant effect of time or the interaction term:

Linear mixed-effects model fit by REML

Data: HL_p53

Random effects:

Formula: ~1 | mouse

(Intercept) Residual

StdDev: 0.1616099 0.06060383

Fixed effects: SVAF ~ tumor_type * weeks

	numDF <int>	denDF <dbl>	F-value <chr>	p-value <chr>
(Intercept)	1	68	586.5492	<.0001

	numDF <int>	denDF <dbl>	F-value <chr>	p-value <chr>
tumor_type	1	68	54.4954	<.0001
Weeks	2	68	1.8302	0.1682
tumor_type:weeks	2	68	0.2742	0.7610

Regarding histological grade, there were only 3 SCCIII tumors, so we elected to bin lesions into well (AK, SCCI) and less differentiated (SCCII, SCCIII) histological groups. A comparison of tumor grade and *Trp53* summed VAF revealed no significant difference between well and less differentiated tumors (2 tailed Wilcoxon test $p=0.89$).

Finally, we examined mixed-effects model on with fixed tumor histology and time and random mouse with the summed VAF for mutant *Trp53* as the response variable:

	numDF	denDF	F-value	P value
(Intercept)	1	68	368.06	<.0001
Histological grade	1	68	0.0101	0.9204
Time (weeks)	2	68	4.9151	0.0190*
Histology:Time	2	68	0.8337	0.4388

*Note that time loses significance ($p=0.057$) when multiple test correction is applied.

In the light of these tests we have moderated our comments on there being two routes of tumor evolution as follows:

Removed the reference to two routes of tumor evolution from the abstract.

Commented that the data is 'consistent' with two evolutionary paths.

'Turning to carcinogenesis, the data is consistent with two convergent evolutionary routes to develop cSCC...' line 402

We have also added the following to the paragraph on the limitations of the study:

'The data is consistent with there being two routes of tumor evolution, *Trp53High* and *Trp53Low*, but a larger sample of lesions would be required to confirm this statistically.'

Lines 463-465.

The comparison of mutant genes in *Trp53*-high vs low tumors (**Fig. 5b**) was corrected for multiple testing using the Benjamini Hochberg method. We have corrected the figure legend to clarify this.

3- The CNA result, that they increase with UV duration and seem to show recurrent chromosomal biases (perhaps selected, although hard to tell) is interesting. I do have some questions about inferring CNA from panel sequencing with a small panel such as here. Ideally we would have validation from (at least shallow) WGS, but it is as it is,

I do not expect them to add this for this study but having extra clarification and perhaps supporting analysis would help.

As mentioned by the reviewer, the use of a small baitset panel is a limitation of the study, Hence a modified version of the QDNAseq framework was used which makes use of off-target reads. These off-target reads should function in a similar way to shallow WGS sequencing, but with a sparser low-coverage distribution.

I understand that the CNA calling mostly draws on off-target reads (fine, there should be plenty of those) however these will have a certain baseline coverage distribution determined by sequence similarities. This does not seem like it will be sufficiently modelled by the G+C content

To confirm that the read counts were not piling up at high homology sequence similarity regions or particular hotspots, the binned reads across all chromosomes (1-19, X, Y). Read counts per bin were counted across samples (both tumour and normal) and plotted in violin plots grouped into 2.5 MB intervals for visibility (Reviewer Fig. 7a, full set of plots is available on Figshare,

https://figshare.com/articles/figure/Plot_of_sequencing_coverage_by_chromosome_used_to_call_copy_number_in_study_of_UV_irradiation_of_SKH1_mice_/30246841?file=58399960) This shows that the binned read counts are generally similar, across the cohort between bins and tumour, normal samples (Reviewer Fig. 7b,c). Typically, there are of the order of 100s of reads per bin.

Reviewer Fig. 6, Supplementary Fig. 5, Copy number analysis using off target reads.

a, Read count of off target reads mapped to Chromosome 1, with genomic start position analysed in 2.5 Mega base intervals, tumors in blue. epidermis in orange. Corresponding plots for all chromosomes are available on Figshare: ([https://figshare.com/articles/figure/Plot of sequencing coverage by chromosome used to call copy number in study of UV irradiation of SKH1 mice /30246841?file=58399960](https://figshare.com/articles/figure/Plot_of_sequencing_coverage_by_chromosome_used_to_call_copy_number_in_study_of_UV_irradiation_of_SKH1_mice_/30246841?file=58399960)) b, c, Dots show median (blue) and maximum (black) read counts by

chromosome in epidermis (**b**) and tumors (**c**). The minimum read count was zero for all chromosomes.

As would be expected there are regions which are relatively enriched in read count, likely due to sequence similarity as correctly suggested by the reviewer. Such increases are seen in both tumour and normal bins as both are captured with the same baitset. As part of the copy number calling pipeline, tumor read counts are normalized against matched normal samples by calculating \log_2 ratios of tumor-to-normal read counts per bin. Because the preferential capture and sequence depth are mirrored, this normalisation largely adjusts for sequence similarities. This is in addition to the corrections using LOESS regression to adjust for GC content and mappability effects.

Due to the conservative settings used for calling the patterns may also appear repetitive and compared to shallow-WGS would likely under-call segments. However given the inherent limitations of this type of sequencing, we submit this is the best attempt to infer CNA that can be made with the available sequencing data.

How is the change in the segmentation coarseness parameter justified? Would the specific CNA calls remain robust after some perturbation of the data (e.g. coverage reduction, sample removal or such)?

The reviewer reasonably asks for justification of the segmentation settings selected. A parameter sweep was performed to quantify how segmentation behaviour changes with different penalty (Segmentation γ) and minimum-size (k-min) settings. We compared five runs ($\gamma = 10$, default $\gamma = 30$, $k_{\min} = 2$, $\gamma = 50$ and $\gamma = 100$) and computed consistent run-level metrics: the number of segments per sample, segment-length distributions (absolute and normalized), and copy-number class composition by summed segment length. These metrics permit direct comparison of segmentation granularity, the relative contribution of different length classes to total segmented bases, and whether copy-number balance (gain/loss/normal proportions) is preserved across parameter choices.

The vast majority of segments are concentrated in very large segments (>50 Mb) for all runs (default $\gamma=30 \rightarrow \sim 89.9\%$; $\gamma=10 \rightarrow \sim 81.6\%$; $\gamma=100 \rightarrow \sim 95.3\%$), indicating that we are conservatively calling chromosome-scale events that dominate the total segmented length (**Reviewer Fig. 6a**). This trend is as expected. Increasing γ leads to a higher segmentation penalty and fewer segments **Reviewer Fig. 6b**). Similarly decreasing the gamma leads to more segments, specifically substantially more short segments (<1 Mb: 304 vs default =169). However, copy-number class composition by summed length is highly stable across runs (for the default run, Gain = 5.36%, Loss = 5.66%, Normal = 88.98%), with only small shifts for other γ values. (**Reviewer Fig. 6c**).

The parameter sweep serves as an explicit robustness check which indicates that chromosome-level CNA patterns are robust to reasonable segmentation-parameter choices for the sequencing data.

The details of the parameter sweep are now included in the methods, as above (Lines 689-708). The code is available at:

https://github.com/PHJonesGroup/Skrupskelyte_etal_SI_code.

We now discuss CNA calling as one of the limitations of the study:

'Limitations of the study follow from the targeted sequencing approach used.... the resolution of the approach used to identify CNA is low and changes smaller than a chromosome scale amplification or deletion may not be detected. The estimate of the proportion of the genome with CNA should be viewed as a lower bound.' Lines 459-462.

Reviewer Figure 7, (Supplementary Fig. 6): Copy number analysis: Results of parameter sweep of segmentation parameter (γ and k -min) settings in copy number analysis **a, Segment length distribution by segmentation parameter. **b**, number of**

segments by segmentation parameter. **c**, Proportion of CNA alterations by segmentation parameter.

Minor:

"BAM files were mapped to the GRCh37d5 reference genome using BWA-mem" this is a human reference genome, not mouse. Copy-paste error? Specify the actual reference used. Also I did not notice mention of the transcript set that was used.

Thank you, the reference genome has been corrected to GRCm38.p6. No transcriptional data was analysed in the study.

Typos: abstract "carinogenesis", spacing in "candidatesfor". These have been corrected.

Response References

1. Martincorena I, *et al.* Universal Patterns of Selection in Cancer and Somatic Tissues. *Cell* **171**, 1029-1041.e1021 (2017).
2. Martincorena I, *et al.* Tumor evolution. High burden and pervasive positive selection of somatic mutations in normal human skin. *Science* **348**, 880-886 (2015).
3. Abascal F, *et al.* Somatic mutation landscapes at single-molecule resolution. *Nature* **593**, 405-410 (2021).
4. Beckman RA. Efficiency of carcinogenesis: Is the mutator phenotype inevitable? *Seminars in Cancer Biology* **20**, 340-352 (2010).
5. Abby E, *et al.* Notch1 mutations drive clonal expansion in normal esophageal epithelium but impair tumor growth. *Nat Genet*, (2023).
6. Martincorena I, *et al.* Somatic mutant clones colonize the human esophagus with age. *Science* **362**, 911-917 (2018).
7. Fowler JC, *et al.* Selection of Oncogenic Mutant Clones in Normal Human Skin Varies with Body Site. *Cancer Discovery* **11**, 340-361 (2021).
8. Herms A, Jones PH. Somatic Mutations in Normal Tissues: New Perspectives on Early Carcinogenesis. *Annual Review of Cancer Biology* **7**, 189-205 (2023).

9. Piipponen M, Riihilä P, Nissinen L, Kähäri V-M. The Role of p53 in Progression of Cutaneous Squamous Cell Carcinoma. *Cancers* **13**, 4507 (2021).
10. King C, *et al.* Somatic mutations in facial skin from countries of contrasting skin cancer risk. *Nature Genetics* **55**, 1440-1447 (2023).
11. Olive KP, *et al.* Mutant p53 gain of function in two mouse models of Li-Fraumeni syndrome. *Cell* **119**, 847-860 (2004).
12. Li YY, Hanna GJ, Laga AC, Haddad RI, Lorch JH, Hammerman PS. Genomic analysis of metastatic cutaneous squamous cell carcinoma. *Clin Cancer Res* **21**, 1447-1456 (2015).
13. Pickering CR, *et al.* Mutational landscape of aggressive cutaneous squamous cell carcinoma. *Clin Cancer Res* **20**, 6582-6592 (2014).
14. Wang NJ, *et al.* Loss-of-function mutations in Notch receptors in cutaneous and lung squamous cell carcinoma. *Proc Natl Acad Sci U S A* **108**, 17761-17766 (2011).
15. Yoshioka A, Nakaoka H, Fukumoto T, Inoue I, Nishigori C, Kunisada M. The landscape of genetic alterations of UVB-induced skin tumors in DNA repair-deficient mice. *Experimental Dermatology* **31**, 1607-1617 (2022).
16. Aiderus A, *et al.* Transposon mutagenesis identifies cooperating genetic drivers during keratinocyte transformation and cutaneous squamous cell carcinoma progression. *PLoS Genet* **17**, e1009094 (2021).
17. van der Weyden L, Alcolea MP, Jones PH, Rust AG, Arends MJ, Adams DJ. Acute sensitivity of the oral mucosa to oncogenic K-ras. *The Journal of Pathology* **224**, 22-32 (2011).
18. Lapouge G, *et al.* Identifying the cellular origin of squamous skin tumors. *Proceedings of the National Academy of Sciences* **108**, 7431-7436 (2011).
19. Bian Y, *et al.* Progressive Tumor Formation in Mice with Conditional Deletion of TGF- β Signaling in Head and Neck Epithelia Is Associated with Activation of the PI3K/Akt Pathway. *Cancer Research* **69**, 5918-5926 (2009).
20. Cammareri P, *et al.* Inactivation of TGF β receptors in stem cells drives cutaneous squamous cell carcinoma. *Nature Communications* **7**, 12493 (2016).

21. Besedina E, Supek F. Copy number losses of oncogenes and gains of tumor suppressor genes generate common driver mutations. *Nature Communications* **15**, 6139 (2024).
22. Weghorn D, Sunyaev S. Bayesian inference of negative and positive selection in human cancers. *Nature Genetics* **49**, 1785-1788 (2017).
23. Tilk S, Tkachenko S, Curtis C, Petrov DA, McFarland CD. Most cancers carry a substantial deleterious load due to Hill-Robertson interference. *eLife* **11**, e67790 (2022).
24. Lawson ARJ, *et al.* Somatic mutation and selection at epidemiological scale. *medRxiv*, 2024.2010.2030.24316422 (2024).
25. Ruiz S, *et al.* Unique and overlapping functions of pRb and p107 in the control of proliferation and differentiation in epidermis. *Development* **131**, 2737-2748 (2004).
26. Ruiz S, Santos M, Lara MF, Segrelles C, Ballestín C, Paramio JsM. Unexpected Roles for pRb in Mouse Skin Carcinogenesis. *Cancer Research* **65**, 9678-9686 (2005).
27. Knudsen ES, Pruitt SC, Hershberger PA, Witkiewicz AK, Goodrich DW. Cell Cycle and Beyond: Exploiting New RB1 Controlled Mechanisms for Cancer Therapy. *Trends in Cancer* **5**, 308-324 (2019).
28. Indra AK, *et al.* Temporally controlled targeted somatic mutagenesis in embryonic surface ectoderm and fetal epidermal keratinocytes unveils two distinct developmental functions of BRG1 in limb morphogenesis and skin barrier formation. *Development* **132**, 4533-4544 (2005).
29. Mardaryev AN, *et al.* p63 and Brg1 control developmentally regulated higher-order chromatin remodelling at the epidermal differentiation complex locus in epidermal progenitor cells. *Development* **141**, 101-111 (2014).
30. Miosge LA, *et al.* Comparison of predicted and actual consequences of missense mutations. *Proc Natl Acad Sci U S A* **112**, E5189-5198 (2015).
31. Solares MJ, Kelly DF. Complete Models of p53 Better Inform the Impact of Hotspot Mutations. *International Journal of Molecular Sciences* **23**, 15267 (2022).

32. Kotler E, *et al.* A Systematic p53 Mutation Library Links Differential Functional Impact to Cancer Mutation Pattern and Evolutionary Conservation. *Molecular Cell* **71**, 178-190.e178 (2018).
33. Kennedy MC, Lowe SW. Mutant p53: it's not all one and the same. *Cell Death & Differentiation* **29**, 983-987 (2022).
34. Murai K, *et al.* Epidermal Tissue Adapts to Restrain Progenitors Carrying Clonal p53 Mutations. *Cell stem cell* **23**, 687-699.e688 (2018).